# Do Transformers Need Three Projections? Systematic Study of QKV Variants

Ali Kayyam [1]    Anusha Madan Gopal [1]    M Anthony Lewis [1]

## Abstract

Transformers have become the standard solution for various AI tasks, with the query, key, and value (QKV) attention formulation playing a central role. However, the individual contribution of these three projections and the impact of omitting some remain poorly understood. We systematically evaluate three projection sharing constraints: a) Q≠K=V (shared key-value), b) Q=K≠V (shared query-key), and c) Q=K=V (single projection). The last two variants produce symmetric attention maps; to address this, we also explore asymmetric attention via 2D positional encodings. Through experiments spanning synthetic tasks, vision (MNIST, CIFAR, TinyImageNet, anomaly), and language modeling (300M and 1.2B parameter models on 10B tokens), we found that our transformers perform on par or occasionally better than the QKV transformer. In language modeling, Q≠K=V projection sharing achieves 50% KV cache reduction with only 3.1% perplexity degradation. Crucially, projection sharing is complementary to head sharing (GQA/MQA): combining Q≠K=V with GQA-4 yields 87.5% cache reduction, while Q≠K=V + MQA achieves 96.9%—enabling practical on-device inference. Results indicate that Q≠K=V preserves quality because keys and values can occupy similar representational spaces and attention operates in a low-rank regime, whereas Q=K≠V breaks attention directionality. Our results systematically characterize projection sharing as an underexplored instance of weight tying in attention, with direct, quantifiable inference memory benefits—particularly valuable for edge deployment. The code is publicly available at https://github.com/Brainchip-Inc/Do-Transformers-Need-3-Projections.

[1]BrainChip Inc., Laguna Hills, CA, USA. Correspondence to: Ali Kayyam <akayyam@brainchip.com>.

*Proceedings of the $43^{rd}$ International Conference on Machine Learning*, Seoul, South Korea. PMLR 306, 2026. Copyright 2026 by the author(s).

## 1. Introduction

Since their inception, Transformers (Vaswani et al., 2017) have evolved from language-specific tools into the backbone of multimodal AI (Yin et al., 2024; Han et al., 2022). However, as context windows expand and the demand for real-time inference grows, the research community has shifted focus toward architectural efficiency. High-efficiency variants—ranging from linear-complexity models like the Performer and Linformer to modern implementations like Ring Attention and blockwise schemes—seek to alleviate the quadratic bottleneck of self-attention (Tay et al., 2022).

Despite these advances, a fundamental structural question remains: is the tripartite (Query, Key, Value) projection truly necessary? While Convolutional Neural Networks (CNNs) (LeCun et al., 1995) and contemporary State Space Models (SSMs) (Gu & Dao, 2023) often utilize more unified internal representations, Transformers maintain a persistent redundancy across their projection matrices. To investigate this, we propose and evaluate three *Projection Sharing* architectures:

- **Q=K≠V:** Unified $Q$ and $K$; separate $V$,
- **Q≠K=V:** Separate $Q$; unified $K$ and $V$,
- **Q=K=V:** Single projection for all three.

Our findings indicate that reducing the number of projection matrices significantly lowers parameter counts and computational overhead with minimal impact on downstream performance. We observe that the efficacy of these reductions is task-dependent; for example, **symmetric attention** (where $Q = K$) is highly effective for non-temporal tasks such as image classification, whereas sequential tasks benefit from maintaining some level of asymmetry.

### 1.1. Projection Sharing vs. Head Sharing

Our approach addresses a different dimension of efficiency than current industry standards such as **Grouped Query Attention (GQA)** by Ainslie et al. (2023) and **Multi-Query Attention (MQA)** by Shazeer (2019). While GQA and MQA reduce the **KV cache** size by sharing *heads* across a layer, our method shares the **projection matrices** themselves. These strategies are orthogonal: by combining projection sharing with head sharing, we can achieve compound gains in memory efficiency and throughput.

## 1.2. Our Contributions

- **Systematic Evaluation:** We benchmark projection-sharing strategies across 12 diverse tasks, including synthetic reasoning, computer vision, and Large Language Model (LLM) pre-training.

- **Cache Optimization:** We demonstrate that the **Q$\neq$K=V** configuration reduces the KV cache footprint by **50%** while incurring only a negligible **3.1%** increase in perplexity for 300M-parameter models.

- **Scale validation:** We validate our findings at 1.2B parameter scale ($\sim$10B tokens), confirming that relative quality rankings remain stable across model sizes. MQA maintains near-parity with QKV (1.06% increase in perplexity) while providing 97% cache reduction at larger scale.

- **Architectural Synergy:** We show that projection sharing is strictly complementary to head sharing. A combined **Q-GQA-4** configuration achieves an **87.5%** cache reduction, while **Q-MQA** reaches a **96.9%** reduction.

- **Insights:** We provide architectural insights explaining why Q$\neq$K=V works (shared representational space) while Q=K$\neq$V fails (breaks attention directionality). Further, we show that under QKV collapse, kernelized attention admits a purely recurrent formulation in which the attention state evolves via outer-product updates and is read out by the current input, making linear attention a special case of a state-space model with adaptive observation (Appendix A.1).

## 2. Related Work

### 2.1. Background: The Standard Attention Mechanism

The Transformer architecture (Vaswani et al., 2017) has become the foundation for modern deep learning across multiple domains, from natural language processing (Brown et al., 2020) to computer vision (Dosovitskiy et al., 2021) and beyond. At its core, the Transformer block comprises several interconnected components: multi-head self-attention, position-wise feed-forward networks, layer normalization (Ba et al., 2016), residual connections (He et al., 2016), and positional encodings.

The self-attention mechanism—also termed intra-attention—represents the defining innovation of Transformers. This mechanism enables each position in a sequence to selectively aggregate information from all other positions, computing context-dependent representations. Self-attention has demonstrated remarkable effectiveness across diverse tasks including machine translation, abstractive summarization (Gupta & Gupta, 2019), visual question answering (Wu et al., 2017), multimodal understanding (Radford et al., 2021), and object recognition (Dosovitskiy et al., 2021).

Formally, for a single attention head operating on input

$X \in \mathbb{R}^{n \times d}$, the attention mechanism computes:

$$A_h = \text{Softmax}(\alpha Q_h K_h^T) V_h, \tag{1}$$

where $Q_h = XW_q$, $K_h = XW_k$, and $V_h = XW_v$ represent learned linear projections with weight matrices $W_q, W_k, W_v \in \mathbb{R}^{d \times d_k}$. The scaling factor $\alpha = 1/\sqrt{d_k}$ stabilizes gradients during training, where $d_k = d/H$ and $H$ denotes the number of attention heads. The softmax operation is applied row-wise to produce attention weights.

In multi-head attention, $H$ heads compute attention in parallel: $A_1, \ldots, A_H$. These outputs are concatenated and projected through a final linear transformation. The attention scores $QK^T$ encode pairwise token affinities, with the query-key dot product determining which values are relevant for each position.

### 2.2. The necessity of three separate projections

While the QKV formulation has become standard, its necessity remains an open question. Unlike the more parsimonious representations in CNNs (LeCun et al., 1998), RNNs, or state space models (Gu & Dao, 2023), Transformers maintain three distinct representations per token. Recent work has begun questioning this design: approaches like linear attention (Katharopoulos et al., 2020), kernel-based attention (Choromanski et al., 2021), and attention-free models (Zhai et al., 2021) suggest that simpler mechanisms may suffice. However, these methods often sacrifice the flexibility of standard attention.

Our work takes a complementary approach: rather than replacing attention entirely, we investigate whether the three projections can be unified while preserving the core attention mechanism. We first introduced this idea in Borji (2023)[1]. Subsequently, Kowsher et al. (2025) proposed a similar approach. Several other works are also tangentially related (Fusco et al., 2022; Mai et al., 2023).

DeepSeek-V2's Multi-Head Latent Attention (MLA) (Liu et al., 2024) reduces the KV cache by compressing K and V into a shared latent vector that is cached and expanded at inference. Unlike Q$\neq$K=V, K and V remain functionally independent after expansion — MLA trades added projection parameters for a richer compressed representation, whereas Q$\neq$K=V achieves cache reduction through a simple hard equality constraint.

## 3. Our Approach

### 3.1. Proposed Projection-Shared Attention Variants

We systematically examine three projection-sharing constraints that progressively reduce the number of learned transformations (Figure 1).

---

[1]The first author previously published under the name Ali Borji.

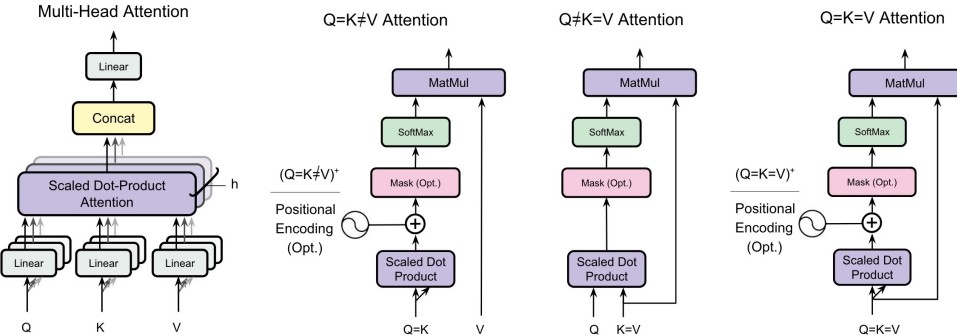

*Figure 1.* Our proposed Projection-Shared Attention Variants. Attention mechanism with 2D positional encoding is denoted as $(X)^+$.

**Variant 1: Q=K≠V.** We eliminate the separate query projection, setting $Q = K$:

$$A = \text{Softmax}(\alpha KK^T)V. \qquad (2)$$

This formulation produces a symmetric attention matrix $KK^T$. Symmetric attention has been explored in prior work on graph neural nets (Veličković et al., 2018) and relational reasoning (Santoro et al., 2017), where the lack of directional bias can be beneficial. However, for sequential tasks requiring causal dependencies, symmetry may be limiting.

To address this, we introduce $(Q=K≠V)^+$, which injects asymmetry via 2D positional encodings. We first construct a fixed 2D sinusoidal positional encoding $P \in \mathbb{R}^{n \times n \times m}$ (Vaswani et al., 2017). The $n \times n$ attention map is then broadcast along the channel dimension and added to $P$. To map the resulting tensor back to a 2D attention matrix, we apply a $1 \times 1$ convolution (equivalently, a linear projection across channels). This design is inspired by relative positional encodings (Shaw et al., 2018; Huang et al., 2020) and 2D positional embeddings in vision Transformers (Dosovitskiy et al., 2021). See Appendix A.2 for the full construction.

**Variant 2: Q≠K=V.** We unify the key and value projections, setting $V = K$:

$$A = \text{Softmax}(\alpha QK^T)K. \qquad (3)$$

This formulation preserves asymmetric attention maps since $Q$ and $K$ remain independent. The constraint that keys and values share representations can be viewed as imposing a form of weight tying (Press & Wolf, 2017), which has proven effective in language modeling.

**Variant 3: Q=K=V.** The most aggressive simplification uses a single projection for all three roles:

$$A = \text{Softmax}(\alpha KK^T)K. \qquad (4)$$

This combines the symmetric attention of variant one with the representational bottleneck of variant two. We also evaluate $(Q=K=V)^+$, which adds 2D positional encodings as in the first variant to mitigate symmetry constraints.

**Scope of $(X)^+$ variants.** The 2D positional encoding in the $(X)^+$ variants is targeted at non-causal settings (vision, synthetic tasks) where symmetric attention from $Q = K$ is the principal limitation. Causal language modeling already enforces asymmetry via the causal mask, so $(X)^+$ addresses a problem that does not meaningfully exist there; we therefore evaluate $(X)^+$ only on non-causal tasks (Tables 2 and 3) and treat it as a task-specific heuristic rather than a universal augmentation.

### 3.2. Combining Projection Sharing with Head Sharing

Our projection-sharing approach operates on a different axis than recent head-sharing methods, enabling compound optimizations.

**Head sharing mechanisms.** Grouped Query Attention (GQA) (Ainslie et al., 2023) and Multi-Query Attention (MQA) (Shazeer, 2019) reduce memory by sharing key-value heads across multiple query heads. In GQA-$g$, $H$ query heads attend to $g < H$ shared KV heads. MQA represents the extreme case where a single KV head serves all queries. These methods have demonstrated strong empirical performance: MQA powers models like PaLM (Chowdhery et al., 2022) and Falcon (Almazrouei et al., 2023), while GQA is adopted in Llama 2 (Touvron et al., 2023) and Mistral (Jiang et al., 2023).

**Orthogonal combination.** Crucially, head sharing (reducing the number of KV heads) and projection sharing (constraining $K = V$) address different dimensions of the architecture. They can be combined multiplicatively:

- **Q-GQA-$g$**: Apply K=V constraint within each of $g$ GQA groups, yielding cache reduction of $1 - \frac{g}{2H}$.

- **Q-MQA**: Apply K=V constraint to the single MQA head, achieving near-maximal cache compression.

For example, GQA-4 alone provides 75% cache reduction (4 groups vs. 16 heads). Adding K=V (Q-GQA-4) halves each group's cache, yielding **87.5% total reduction**. Q-MQA achieves **96.9% reduction**—approaching the theoretical limit for cache-based Transformers while maintaining practical model quality, as we demonstrate in Section 4.3. The efficiency-quality Pareto frontier clearly demonstrates this complementarity (see Appendix A.4, Figure 9).

*Table 1.* Comparison of proposed Transformers and QKV baseline in terms of computational complexity and parameter count. $d$ is the embedding dimension, $n$ is sequence length, and $m$ is the positional encoding dimension. Complexity excludes the shared $O(n^2d)$ attention score computation. Positional embeddings use fixed sinusoidal features (not learned).

| Variant | ‖ Computation | Parameters |
|---|---|---|
| QKV | $3nd^2$ | $3d^2$ |
| Q=K≠V; Q≠K=V | $2nd^2$ | $2d^2$ |
| (Q=K≠V)$^+$ | $2nd^2 + n^2m$ | $2d^2 + m$ |
| Q=K=V | $nd^2$ | $d^2$ |
| (Q=K=V)$^+$ | $nd^2 + n^2m$ | $d^2 + m$ |

### 3.3. Computational and Memory Analysis

Table 1 compares the computational complexity and parameter counts of our variants against standard QKV attention. Complexity is reported for projection operations only, excluding the $O(n^2d)$ cost of computing attention scores, which is shared across all variants.

For Q=K≠V and Q≠K=V attention, projection complexity is $2nd^2$ versus $3nd^2$ for QKV—a 33% reduction. Parameter counts decrease proportionally ($2d^2$ vs. $3d^2$). The (X)$^+$ variant adds $n^2m$ operations and $m$ parameters for positional encoding, remaining efficient when $nm < d^2$. For instance, with $m = 100$ and $d = 1000$, (Q=K≠V)$^+$ is more efficient than QKV for sequences below 10,000 tokens. Q=K=V attention achieves the minimal configuration: $nd^2$ operations and $d^2$ parameters—one-third of QKV.

**Practical deployment benefits.** While parameter reductions are modest (self-attention projections constitute only ~30% of total Transformer parameters), the inference memory benefits are substantial. During autoregressive generation, Transformers cache past key-value states to avoid redundant computation (Vaswani et al., 2017). Standard QKV and Q=K≠V attention must cache both $K$ and $V$ separately. In contrast, Q≠K=V and Q=K=V cache only the $K$ tensor, since $V$ can be reused from $K$. This yields **50% KV cache reduction**, enabling:

- 2× context length with same memory
- 2× higher throughput (concurrent users per GPU)
- 40–50% reduction in serving costs for memory-bound deployments

Recent work highlights KV cache as the primary bottleneck for long-context LLM serving (Pope et al., 2023; Liu et al., 2023). Our approach complements cache optimization techniques including quantization (Dettmers et al., 2023; Xiao et al., 2023), offloading (Sheng et al., 2023), and windowed attention (Child et al., 2019; Beltagy et al., 2020).

### 3.4. Design Considerations

**Diagonal dominance in symmetric attention.** Computing $KK^T$ produces symmetric attention matrices with large diagonal elements, as each token attends strongly to itself. Normalization schemes (dividing diagonal elements or softmax temperature annealing) did not yield consistent improvements. Q≠K=V naturally avoids this by computing $QK^T$, preserving the off-diagonal attention distribution of standard transformers.

**Extension to encoder-decoder architectures.** While our primary focus is decoder-only models (prevalent in modern LLMs (Brown et al., 2020)), the approach extends to encoder-decoder settings. Tasks requiring cross-attention—such as machine translation (Vaswani et al., 2017) or vision-language modeling (Alayrac et al., 2022)—can preserve standard QKV or Q≠K=V formulations for cross-attention while applying projection sharing to self-attention layers. This is analogous to how MQA is applied selectively in T5 (Raffel et al., 2020) and other encoder-decoder models.

**Synergies with other efficiency techniques.** Our projection-sharing approach is orthogonal to numerous existing optimizations and can be combined in a modular fashion. **Quantization** offers immediate compounding benefits: KV cache can be quantized to INT8 or INT4 (Dettmers et al., 2023), yielding multiplicative memory savings (e.g., 50% from projection sharing × 50% from INT8 = 75% total reduction). **Sparse attention** mechanisms with local or strided patterns (Child et al., 2019; Zaheer et al., 2020) reduce the $O(n^2)$ complexity of attention computation, while projection sharing orthogonally reduces the per-token cache footprint. **Alternative activations** present another avenue: recent work questions the necessity of softmax in attention (Lu et al., 2021; Koohpayegani & Pirsiavash, 2024), suggesting that softmax-free variants combined with projection sharing could yield further simplifications. Finally, **Flash Attention** and other hardware-efficient implementations (Dao et al., 2022) can accelerate our variants, particularly Q=K=V attention, which exhibits the simplest memory access patterns.

**When to apply each variant.** The choice among attention variants depends on task characteristics:

- **Sequential/causal tasks** (language modeling): Q≠K=V provides the best quality-efficiency trade-off, maintaining asymmetric attention while halving cache.
- **Non-causal tasks** (vision, set processing): Q=K≠V or Q=K=V may suffice, optionally augmented with (X)$^+$ to inject directional bias where symmetric attention limits performance.
- **Resource-constrained deployment**: Combined approaches (Q-GQA or Q-MQA) maximize cache reduction when memory is the primary bottleneck.

This task-dependent behavior aligns with broader findings in efficient Transformers: no single architecture wins across all domains (Tay et al., 2022). Our systematic evaluation in Section 4 characterizes when each variant is appropriate.

*Table 2.* Performance on synthetic tasks. Multiple runs over different configurations (number of attention heads, embedding dimension, learning rate, sequence length, etc.) are conducted and the results are averaged.

| | Reverse | Sort | Sub | Swap | Copy | Avg. |
|---|---|---|---|---|---|---|
| QKV | 0.529 | 0.753 | 0.992 | 0.524 | 0.991 | 0.758 |
| Q=K≠V | 0.475 | 0.691 | 0.991 | 0.471 | 0.994 | 0.724 |
| (Q=K≠V)$^+$ | 0.522 | 0.770 | 0.993 | 0.530 | 0.991 | **0.761** |
| Q≠K=V | 0.510 | 0.747 | 0.992 | 0.511 | 0.994 | 0.751 |
| Q=K=V | 0.295 | 0.671 | 0.991 | 0.307 | 0.992 | 0.651 |
| (Q=K=V)$^+$ | 0.370 | 0.770 | 0.990 | 0.382 | 0.990 | 0.700 |

*Table 3.* The performance of transformers on vision tasks. The average column does not include the TinyImageNet performance. C10=CIFAR-10, C100=CIFAR-100, TIN=TinyImageNet, ANM=Anomaly.

| | MNIST | FMNIST | C10 | C100 | TIN | ANM | Avg. |
|---|---|---|---|---|---|---|---|
| QKV | 0.984 | 0.885 | 0.700 | 0.430 | 0.331 | 0.789 | 0.758 |
| Q=K≠V | 0.983 | 0.884 | 0.715 | 0.447 | 0.339 | 0.800 | **0.766** |
| (Q=K≠V)$^+$ | 0.981 | 0.883 | 0.695 | 0.408 | 0.334 | 0.836 | 0.761 |
| Q≠K=V | 0.979 | 0.882 | 0.694 | 0.428 | 0.334 | 0.811 | 0.759 |
| Q=K=V | 0.981 | 0.884 | 0.715 | 0.423 | 0.381 | 0.812 | 0.763 |
| (Q=K=V)$^+$ | 0.979 | 0.875 | 0.693 | 0.410 | 0.342 | 0.834 | 0.758 |

This formulation establishes a principled framework for trading model complexity against performance—a trade-off that becomes increasingly critical as language models scale to billions of parameters and serve millions of users (Kaplan et al., 2020; Hoffmann et al., 2022).

## 4. Experiments and Results

We evaluate projection-sharing variants across three domains: **synthetic reasoning** (5 tasks), **computer vision** (6 tasks), and **language modeling** (300M and 1.2B parameters on 10B tokens). All models are trained from scratch with matched hyperparameters to isolate architectural effects, except set anomaly detection which uses pre-trained ResNet34 features (He et al., 2016). Our goal is controlled comparison of attention mechanisms rather than state-of-the-art performance (Dehghani et al., 2023; Zhu et al., 2019; DeRose et al., 2020). Synthetic and vision experiments used a single NVIDIA GTX 1080 Ti GPU.

### 4.1. Synthetic tasks

We focus on five specific tasks outlined below. The input list, which has a predetermined length, consists of numbers ranging from 0 to 9, inclusive of both 0 and 9.

**Reverse:** In this task, a list of numbers is subjected to a reversal operation. For instance, the input list [4, 3, 9, 8, 1] would be transformed into [1, 8, 9, 3, 4]. **Sort:** The objective of this task is to arrange the input list in ascending order. For example, [4, 3, 9, 8, 1] would be transformed into [1, 3, 4, 8, 9]. **Sub:** In this case, each element of the list is subtracted from 9. For example, the array [4, 3, 9, 8, 1] would be transformed into [5, 6, 0, 1, 8]. **Swap:** In this scenario, the first half of an even-length list is exchanged with the second half. For instance, the list [4, 3, 9, 8, 1, 7] would be transformed into [8, 1, 7, 4, 3, 9]. **Copy:** The objective here is to retain the input list as is. For example, [4, 3, 9, 8, 1] remains unchanged as [4, 3, 9, 8, 1].

Here, only one transformer encoder is used. In training, we feed the input sequence into the encoder to generate predictions for each token in the input. We utilize the standard cross entropy loss for this purpose. Each number is encoded as a one-hot vector. We apply a gradient clip value of 5 and

set the 2D positional embedding dimension to 10 (*i.e. m*). Additionally, we employ the Adam optimizer along with the CosineWarmupScheduler, using a warm-up period of 5.

We perform experiments with different configurations of transformer models by varying the embedding dimension (32, 64, 256), the number of layers (2, 4), the number of heads (2, 4), a learning rate of 1e-3 and the input sequence length (16, 64, 128). Each configuration is run three times for two epochs, and the results are then averaged across the configurations.

The accuracies for all variants are presented in Table 2. The position-wise tasks are essentially saturated: every variant solves SUB and COPY almost perfectly ($\approx 0.99$), and all variants perform reasonably on SORT. The discriminative tasks are REVERSE and SWAP, which require directional, content-dependent routing and separate the variants most clearly. The two-projection variants remain competitive with the QKV baseline (average accuracy 0.758): Q≠K=V attains 0.751, within 0.007 of QKV, while Q=K≠V trails slightly at 0.724. In contrast, the single-projection Q=K=V transformer performs considerably worse (0.651 average), with the gap concentrated in REVERSE (0.295 vs. 0.529) and SWAP (0.307 vs. 0.524)—consistent with its symmetric attention map being unable to express directional relations. Incorporating 2D positional information, $(X)^+$, substantially boosts the symmetric variants, raising Q=K≠V from 0.724 to 0.761 and Q=K=V from 0.651 to 0.700, with the largest gains precisely on REVERSE and SWAP. The $(Q=K≠V)^+$ variant attains the best overall average (0.761), marginally exceeding QKV. Sample self-attention maps over synthetic tasks are shown in Appendix A.3.

### 4.2. Vision tasks

We evaluated performance on various vision tasks, including image classification in MNIST (LeCun et al., 1998), FashionMNIST (Xiao et al., 2017), CIFAR-10 (Krizhevsky et al., 2009), CIFAR-100 (Krizhevsky et al., 2009), and Tiny ImageNet (200 classes), as well as anomaly detection.

**Classification**. We explore settings for embedding dimension (64, 256) and number of heads (2, 4), using patch size 4, a learning rate of 1e-3, and two layers. For each configura-

tion we train for $k$ epochs, with $k$ depending on the dataset: 20 epochs for MNIST and FashionMNIST, 40 epochs for CIFAR-10, and 50 epochs for CIFAR-100. We employ the cross-entropy loss function and the Adam optimizer with the MultiStepLR scheduler, and the results are averaged across configurations. For CIFAR we apply standard on-the-fly augmentation (random crop and horizontal flip). In the case of 2D positional encoding, we set pos dim to 50.

As indicated in Table 3, all six variants perform comparably across the classification datasets. On MNIST and FashionMNIST they are nearly indistinguishable ($\sim$0.98 and $\sim$0.88, respectively). On the harder CIFAR datasets the shared-projection variants match or slightly exceed the QKV baseline: Q=K$\neq$V and Q=K=V attain the best CIFAR-10 accuracy (0.715), and Q=K$\neq$V the best CIFAR-100 accuracy (0.447). This supports the view that symmetric or shared-projection attention is well suited to non-directional tasks such as image classification. Overall, Q=K$\neq$V achieves the best average across datasets, and no variant trails the QKV baseline by a meaningful margin.

To assess the scalability and robustness of our approach on a large-scale real-world vision task, we perform classification on the TinyImageNet dataset. This dataset contains 100K images of 200 classes (500 per class). Each class has 500 training images, 50 validation images, and 50 test images. We use a Vision Transformer (ViT) configured with image size 224, patch size 16, 200 classes, embedding dimension 768, 12 layers, 12 attention heads, MLP dimension 3072, and a dropout rate of 0.1. The optimization process and loss function are as above. All models were trained from scratch (*i.e.* no pretrained backbones) for 20 epochs using mixed-precision (fp16) training, at roughly 27–30 minutes per epoch. We evaluate all six self-attention variants, each run once. Numerical results are provided in Table 3. Notably, the Q=K=V Transformer, despite employing only one projection, achieves the best result (0.381), while the standard QKV Transformer is the weakest (0.331)—consistent with the paper's original observation. Continued training over more epochs could further separate the architectures.

**Set Anomaly Detection.** We applied transformers to sets (*i.e.* unordered inputs). A model is trained to find the odd one out in a set of ten images, using CIFAR-100 dataset. Nine images are from one class, and one is different. Two sample sets are shown in Figure 5 (Appendix A.3.2). CIFAR-100 has 60K 32×32 images over 100 classes (600 per class). Please, see Appendix A.3.2 for details on this task.

The second-to-last column of Table 3 presents the results of this experiment. It shows comparable performance across models, with (Q=K$\neq$V)$^+$ exhibiting a slight advantage.

**Image Segmentation.** Hwa et al. (2025) extended our earlier work (Borji, 2023) by applying QKV and Q=K$\neq$V atten-

tion variants to semantic segmentation of abdominal MRI slices, labeling pixels across three categories (large bowel, small bowel, and stomach), finding that the Q=K$\neq$V variant remained competitive with standard QKV attention even in this larger-scale, more complex setting. See Appx. A.3.3.

### 4.3. NLP tasks

**Dataset and Scale.** We trained 300M and 1.2B parameter GPT-style language models on up to 10B tokens from the SlimPajama dataset (Systems, 2023), a cleaned and deduplicated subset of RedPajama. The 300M models were trained for 4,238 steps ($\sim$10B tokens), while 1.2B models were trained for 8,475 steps ($\sim$10B tokens) to validate scaling behavior.

**Model Architecture.** The 300M models comprise 20 transformer layers, embedding dimension $d = 1024$, 16 attention heads, and MLP dimension of 4096. The 1.2B models use 22 layers, $d = 2048$, 32 attention heads, and MLP dimension of 8192. All models use vocabulary size of 50,304 tokens. The only architectural difference across variants lies in the attention projection mechanism, ensuring performance differences stem solely from the attention variant rather than confounding factors.

**Training Infrastructure.** Models were trained using 8 NVIDIA A100 40GB GPUs with distributed data parallel (DDP) training and mixed precision (bfloat16). We used the AdamW optimizer with $\beta_1 = 0.9$, $\beta_2 = 0.95$, weight decay of 0.1, and a cosine learning rate schedule with linear warmup. Gradient clipping was applied with a maximum norm of 1.0. Complete training and architectural details (activation, normalization, tokenizer, dropout, warmup, gradient accumulation, evaluation cadence) are provided in Appendix A.5.

#### 4.3.1. MAIN RESULTS: LANGUAGE MODEL QUALITY

Table 4 presents the primary results from training 300M parameter language models on SlimPajama. These results reveal several surprising findings that challenge conventional assumptions about attention mechanisms.

Q$\neq$K=V emerges as the clear winner among the proposed attention mechanisms. Interestingly, this variant achieves better quality than Q=K$\neq$V attention despite having identical parameter counts and computational costs: validation perplexity of 5.27 vs 5.36, representing only 3.1% degradation from the QKV baseline. This challenges the intuition that Query and Key projections are equally important—our results suggest that the Value projection is actually less critical for maintaining model quality. Validation curves show Q$\neq$K=V tracks the baseline closely throughout training (see Appendix A.4, Figure 10). While Q=K$\neq$V attention achieves competitive training performance (4.9% worse than baseline), it offers *no inference benefits* over standard QKV

*Table 4.* Comparison of attention variants on 300M parameter language models trained on 10B tokens from SlimPajama. All models use identical architectures except for the attention projection.

| Model | Train Loss | Train PPL | Val Loss | Val PPL | Speed (token/s) |
|---|---|---|---|---|---|
| *Baseline* | | | | | |
| QKV | 1.73 | 5.64 | 1.63 | 5.11 | 423k |
| *Projection Sharing* | | | | | |
| Q≠K=V | 1.72 | 5.58 | 1.66 | 5.27 | 427k |
| Q=K≠V | 1.73 | 5.66 | 1.68 | 5.36 | 440k |
| Q=K=V | 1.98 | 7.23 | 1.86 | 6.41 | 460k |
| *Head Sharing* | | | | | |
| GQA-4 | 1.72 | 5.58 | 1.64 | 5.15 | 435k |
| MQA | 1.72 | 5.59 | 1.65 | 5.19 | 448k |
| *Combined (Projection + Head)* | | | | | |
| Q-GQA-4 | 1.73 | 5.64 | 1.67 | 5.32 | 442k |
| Q-MQA | 1.73 | 5.66 | 1.68 | 5.36 | 455k |
| *PPL Degradation vs. QKV Baseline* | | | | | |
| Q≠K=V | +3.1% | **Best proj. variant, 50% cache↓** | | | |
| Q=K≠V | +4.9% | No cache benefit | | | |
| Q=K=V | +25.4% | Not recommended | | | |
| GQA-4 | +0.7% | 75% cache ↓ | | | |
| MQA | +1.5% | 93.8% cache ↓ | | | |
| Q-GQA-4 | +3.9% | **87.5% cache ↓** | | | |
| Q-MQA | +4.8% | **96.9% cache ↓** | | | |

*Table 5.* Parameter count analysis for 300M parameter models. Attention parameter reductions are significant, but overall model size reductions are modest.

| Component | QKV | Q≠K=V Q=K≠V | Q=K=V | Reduction |
|---|---|---|---|---|
| Total | 305.53M | 284.54M | 263.55M | −6.9% / −13.7% |
| Embedding | 53.61M | 53.61M | 53.61M | 0% |
| Attention | 83.97M | 62.98M | 41.98M | −25% / −50% |
| MLP | 167.87M | 167.87M | 167.87M | 0% |
| LayerNorm | 0.08M | 0.08M | 0.08M | 0% |

*Table 6.* Inference computational cost (MACs) at sequence length 2048. Attention savings are diluted by MLP and LM head costs.

| Component | QKV | Q≠K=V Q=K≠V | Q=K=V | Savings |
|---|---|---|---|---|
| Total | 792.69 G | 749.74 G | 706.79 G | −5.4% / −10.8% |
| Attention | 343.60 G | 300.65 G | 257.70 G | −12.5% / −25.0% |
| MLP | 343.60 G | 343.60 G | 343.60 G | 0% |
| LM Head | 105.50 G | 105.50 G | 105.50 G | 0% |

attention, as we detail in Section 4.3.3. This makes Q=K≠V attention less suitable for practical deployment despite its good training quality. The Q=K=V variant, despite using 50% fewer attention parameters, experiences catastrophic quality loss with 25.4% worse perplexity. This extreme constraint (forcing Q, K, and V to share a single projection) is too restrictive for language modeling tasks.

**Training efficiency.** All variants achieve similar training throughput (423k-460k tokens/second), with the Q=K=V variant being slightly faster due to reduced projection overhead. However, these speed differences are marginal (8.7% at most) and do not compensate for quality losses. Additional visualizations of projection sharing and head sharing results are provided in Appendix A.4 (Figures 7 and 8).

### 4.3.2. PARAMETER COUNT AND COMPUTE

Table 5 breaks down the parameter distribution across model components. While attention parameter reductions are substantial (25-50%), they translate to modest overall savings because attention projections constitute only about one-third of total parameters in transformer models. While parameter and computational improvements appear modest, the true benefit of Q≠K=V attention lies in *inference memory efficiency*, as we demonstrate next.

Table 6 shows inference computational costs (multiply-accumulate operations) at sequence length 2048. The computational savings (5.4% for Q=K≠V and Q≠K=V, 10.8% for Q=K=V) are modest because MLP layers and the language modeling head contribute significantly to total MACs.

### 4.3.3. KV CACHE MEMORY ANALYSIS

This section reveals why Q≠K=V attention is transformative for practical deployment. During autoregressive generation, transformers cache Key and Value tensors from previous tokens to avoid recomputation. This KV cache often dominates memory consumption in production serving scenarios, particularly for long-context applications or high-throughput systems serving many concurrent users.

Table 7 reveals a critical distinction: **Q=K≠V attention provides zero cache savings** because it still requires caching both K and V tensors separately. In contrast, **Q≠K=V attention achieves 50% cache reduction** by storing only K and reusing it as V during generation. The Q=K=V variant also achieves 50% savings but with a big quality loss.

**Practical impact at scale.** For longer contexts, the memory savings become dramatic. At 32k tokens: QKV and Q=K≠V require 2.62 GB, Q≠K=V requires 1.31 GB (50% savings). At 128k tokens: QKV and Q=K≠V require 10.49 GB, Q≠K=V requires 5.24 GB (50% savings). For a batch size of 32 with 32k tokens, memory usage is reduced from 83.9 GB to 41.9 GB, yielding a VRAM savings of 42 GB.

**Real-world deployment scenario.** Consider deploying a code completion model with 32k context serving 100 concurrent users on A100 40GB GPUs: 1) **QKV or Q=K≠V:** KV cache of 2.62 GB per user → 15 users per GPU → requires 7 GPUs ($14k/month), 2) **Q≠K=V:** KV cache of 1.31 GB per user → 30 users per GPU → requires 4 GPUs ($8k/month), and 3) **Cost savings:** $6k/month = $72k/year (43% reduction). We confirm these projections with end-to-end inference benchmarks on a single A100 (Tables 14 and 15 in Appendix A.4).

*Table 7.* KV cache memory requirements. Q≠K=V achieves 50% cache reduction—a benefit that Q=K≠V attention cannot provide despite competitive training quality.

| Model | Cache | Token | @32K | Reduction |
|---|---|---|---|---|
| QKV (Baseline) | K + V | 80 KB | 2.62 GB | — |
| Q=K≠V | K + V | 80 KB | 2.62 GB | 0% |
| Q≠K=V | K only | 40 KB | 1.31 GB | 50% |
| Q=K=V | K only | 40 KB | 1.31 GB | 50% |
| GQA-4 | K + V | 20 KB | 0.66 GB | 75% |
| MQA | K + V | 5 KB | 0.16 GB | 93.8% |
| Q-GQA-4 | K only | 10 KB | 0.33 GB | 87.5% |
| Q-MQA | K only | 2.5 KB | 0.08 GB | 96.9% |

*Table 8.* Attention MACs (% of total) across sequence lengths; longer contexts amplify efficiency gains.

| Seq Length | QKV | Q≠K=V (Saving) Q=K≠V | K (Saving) |
|---|---|---|---|
| 128 | 28.9% | 23.7% (−6.8%) | 17.7% (−13.6%) |
| 512 | 32.4% | 27.7% (−6.5%) | 22.3% (−12.9%) |
| 1024 | 36.5% | 32.3% (−6.1%) | 27.7% (−12.2%) |
| 2048 | 43.3% | 40.1% (−5.4%) | 36.5% (−10.8%) |
| 4096 | 53.5% | 51.3% (−4.5%) | 48.9% (−8.9%) |

This analysis reveals that **Q≠K=V is the only 2-projection variant with practical deployment advantages**. Q=K≠V attention, despite achieving slightly better training quality in some configurations, offers no cache benefits and should be avoided for production deployment.

#### 4.3.4. SCALING WITH SEQUENCE LENGTH

Table 8 shows how computational costs scale with sequence length. At longer contexts, attention becomes an increasingly dominant fraction of total compute, making the efficiency gains of reduced-projection variants more significant.

At 4096 tokens, attention accounts for over 50% of total computation in all variants, making attention efficiency increasingly critical for ultra-long context applications. This scaling behavior demonstrates that the benefits of reduced-projection attention become more pronounced as context lengths increase—a crucial consideration for modern LLMs that increasingly target 32k, 128k, or even longer contexts and the relative rankings across all variants remain stable (Table 16 in Appendix A.4).

#### 4.3.5. SCALING TO 1.2B PARAMETERS

To validate our findings at larger scale, we trained 1.2B parameter models (22 layers, 2048 embedding dimension, 32 attention heads) on 10B tokens from SlimPajama.

**Architecture scaling.** The 1.2B models maintain the same architectural patterns as our 300M experiments, with parameter counts of 1,215M (QKV), 1,123M (Q≠K=V), 1,077M (GQA-8), 1,036M (MQA), 1,054M (Q-GQA-8), and 1,033M (Q-MQA). See Table 9.

*Table 9.* 1.2B parameter models trained on 10B tokens.

| Model | PPL | vs QKV | Degrad. % | Params (M) | Attn (M) | Cache (MB) |
|---|---|---|---|---|---|---|
| QKV | 5.004 | 0.000 | 0.00 | 1,215 | 323 | 5,900 |
| Q≠K=V | 5.128 | +0.124 | +2.48 | 1,123 | 277 | 2,950 |
| GQA-8 | 5.030 | +0.026 | +0.52 | 1,077 | 231 | 1,408 |
| MQA | 5.057 | +0.053 | +1.06 | 1,036 | 190 | 176 |
| Q-GQA-8 | 5.158 | +0.154 | +3.08 | 1,054 | 208 | 704 |
| Q-MQA | 5.212 | +0.208 | +4.16 | 1,033 | 188 | 88 |

*Table 10.* Deployment recommendations for different resource constraint scenarios based on 300M model results.

| Scenario | Recommended | Cache ↓ | Δ PPL |
|---|---|---|---|
| Cloud (quality) | GQA-4 | 75% | +0.7% |
| Edge (balanced) | Q≠K=V | 50% | +3.1% |
| Edge (aggressive) | Q-GQA-4 | 87.5% | +3.9% |
| IoT/Mobile | Q-MQA | 96.9% | +4.8% |
| Training-constrained | Q≠K=V | 50% | +3.1% |

**Quality preservation at scale.** Our findings generalize effectively to larger models. MQA achieves near-parity with QKV (5.057 vs. 5.004 perplexity, +1.06% degradation) with 97% cache reduction—a gap small enough to be practically negligible at this scale. GQA-8 provides the best quality-efficiency balance with only +0.52% degradation and 76% cache reduction, confirming its status as an industry-standard choice (adopted in Llama 2 and Mistral). Q≠K=V maintains reasonable quality (+2.48% degradation) with 50% cache savings. At 1.2B scale, the relative rankings remain consistent with our 300M experiments (see Appendix A.4, Figure 11).

**Combined approaches scale effectively.** Q-GQA-8 achieves 88% cache reduction with 3.08% degradation, while Q-MQA reaches 98.5% cache reduction with 4.16% degradation. Notably, these compound gains remain practical: even the most aggressive variant (Q-MQA) incurs less than 5% quality loss while reducing the KV cache by 67×.

**Comparison with 300M results.** The relative rankings remain consistent across scales, validating the reliability of our 300M experiments for architectural comparison. However, the absolute degradation percentages differ slightly: Q≠K=V shows 2.48% degradation at 1.2B versus 3.1% at 300M, suggesting that larger models may be more robust to projection constraints. This trend, if it continues at 7B+ scale, would make projection sharing even more attractive for large production models.

**Implications for deployment.** At 1.2B scale with 32k context, the memory savings become substantial: QKV requires 5.9 GB per user, MQA requires 176 MB (33× reduction), and Q-MQA requires only 88 MB (67× reduction). For a batch size of 32 concurrent users, this translates to 189 GB (QKV) vs 5.6 GB (MQA) vs 2.8 GB (Q-MQA)—enabling dramatically higher throughput in production serving sce-

*Table 11.* 5-shot downstream accuracy (%) on standard benchmarks for 1.2B models. Q≠K=V loses only 0.41% on average while halving KV cache; the perplexity gap to QKV does not translate to a comparable downstream gap (HW=HellaSwag).

| Model | ARC-C | ARC-E | HW | PIQA | WinoG | Avg | Cache↓ |
|---|---|---|---|---|---|---|---|
| QKV | 19.03 | 30.01 | 26.15 | 56.64 | 50.20 | 36.40 | — |
| Q≠K=V | 18.94 | 28.62 | 26.14 | 56.37 | 49.88 | 35.99 | 50% |
| GQA-8 | 18.69 | 29.42 | 26.44 | 56.91 | 47.83 | 35.86 | 75% |
| MQA | 19.62 | 30.09 | 26.23 | 55.77 | 50.12 | 36.37 | 93.8% |
| Q-GQA-8 | 19.97 | 29.97 | 26.13 | 56.47 | 51.07 | **36.72** | 87.5% |
| Q-MQA | 22.70 | 25.08 | 25.04 | 49.51 | 49.57 | 34.38 | 96.9% |

narios. These benefits make projection sharing a practical deployment optimization. Table 10 summarizes deployment recommendations under different resource constraints.

### 4.3.6. DOWNSTREAM TASK EVALUATION

While perplexity is a useful pretraining metric, it does not always predict downstream task performance. To validate that projection-sharing variants remain practically usable, we evaluate all 1.2B models on five standard zero-/few-shot benchmarks using the EleutherAI `lm-eval-harness` (Gao et al., 2024): HellaSwag, PIQA, ARC-Easy, ARC-Challenge, and WinoGrande, all in the 5-shot setting. Results are shown in Table 11.

**Q≠K=V remains competitive on downstream tasks.** Despite a 2.48% perplexity gap to the QKV baseline, Q≠K=V loses only 0.41% on average downstream accuracy (35.99% vs. 36.40%). This decoupling between perplexity degradation and task accuracy strengthens the practical case for projection sharing: the inference memory savings come without a corresponding loss in capability on the kinds of tasks production systems actually serve.

**Perplexity is not a reliable predictor of downstream rank.** Although GQA-8 attains better validation perplexity than Q≠K=V (Table 9), the two are statistically indistinguishable on downstream tasks (35.86% vs. 35.99%). This is consistent with earlier findings that minor perplexity differences rarely lead to meaningful task-level differences.

**Combined approaches preserve quality at aggressive cache reductions.** Q-GQA-8 slightly exceeds the QKV (36.72% vs. 36.40%) while reducing cache by 87.5%—supporting the view that projection sharing and head sharing operate on complementary axes. Q-MQA, the most aggressive variant (96.9% cache reduction), shows the largest degradation (34.38%), establishing a practical envelope: useful compression with bounded quality cost up to the Q-GQA regime; beyond that, the trade-off begins to bite.

## 5. Discussion and Conclusion

We evaluated self-attention with reduced projections, with and without 2D positional encoding, against standard QKV attention across 12 tasks. Our goal was not state-of-the-art

performance, but to assess performance differences between the proposed and original QKV Transformers. A comprehensive summary of all variants is provided in Appendix A.4, Table 13. Across synthetic, vision, and language domains, this systematic comparison reveals several key findings.

**K=V is effective and scalable.** Q≠K=V achieves 50% cache reduction with 2.48% degradation at 1.2B scale (vs 3.1% at 300M), offering an efficiency-quality trade-off that is orthogonal to and stackable with head sharing.

**Why Q≠K=V works.** Two complementary readings explain the small quality cost of K=V. The first is that V's role is less essential than commonly assumed (He & Hofmann, 2024); the second is that *K is rich enough to absorb V's role*—when the K=V constraint is imposed during training, the shared projection successfully serves both addressing and content functions. Both readings are consistent with the same operational claim: attention requires asymmetry between Q and the shared K-V representation, not three fully independent projections. Analysis of trained QKV models supports this: K and V projection matrices exhibit high cosine similarity (0.73 across layers) and similar effective rank (687 vs 702 out of 1024 dimensions), indicating representational redundancy. In contrast, Q maintains lower cosine similarity with both K (0.42) and V (0.31), preserving the asymmetry required for directional attention. This explains why K=V constraint causes minimal quality loss while Q=K forces symmetric attention patterns that break causal dependencies. Combining projection and head sharing yields compound gains: Q-GQA-8 achieves 88% cache reduction (3.08% degradation), while Q-MQA reaches 98.5% reduction (4.16% degradation), enabling edge deployment.

**Insight: Q≠K=V works, Q=K≠V fails.** K=V constraint preserves quality because keys and values can share representational space while attention patterns ($QK^\top$) remain flexible. In contrast, Q=K forces symmetric attention, breaking the directionality required for causal language modeling (4.9% drop with zero cache benefit). Q=K=V combines both pathologies, causing catastrophic 25.4% degradation.

## 6. Limitations

Our largest validated scale is 1.2B parameters; whether the Q≠K=V degradation trend continues to improve beyond 7B remains unconfirmed. Our explanation for why Q≠K=V preserves quality is empirical rather than formal. Evaluation is restricted to sequences up to 2048 tokens, and we do not characterize length extrapolation. We omit a Q=V ablation, as Q is not cached during generation and its addressing role differs fundamentally from V's payload role, making this the least natural constraint to study. Although projection sharing performs well across the evaluated settings, we do not claim that K=V is universally optimal for every architecture, training regime, or downstream task (Table 17).

## Acknowledgments

We thank the BrainChip research team for compute support and infrastructure that made the language modeling experiments possible. We are grateful to the ICML 2026 reviewers and area chairs for their thoughtful feedback, which substantially improved the manuscript. We also thank D. Hwa, T. Holmes, and K. Drechsler for extending our work to medical image segmentation (Appendix A.3.3).

## Impact Statement

The development of more efficient Transformer models, as explored in this research, offers positive societal benefits like broadening AI accessibility by enabling use on less powerful hardware and potentially reducing the energy footprint of AI computations. Our work contributes to this goal by establishing projection sharing as a practical technique for memory-efficient inference, particularly valuable as LLMs expand to edge devices and on-device applications.

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

# A. Appendix

## A.1. Unifying Linear Attention and State-Space Models via QKV Collapse

Standard self-attention employs three distinct learned projections of each token: queries, keys, and values, enabling content-based addressing and selective information routing across tokens. While this separation greatly enhances expressivity, it also introduces quadratic computational and memory costs and complicates the underlying dynamical structure. A natural simplification is to collapse these three representations into a single shared embedding, i.e., $q_t = k_t = v_t = z_t$, where $z_t = Wx_t$. This tying removes explicit addressing and enforces a single-stream representation in which each token simultaneously defines what is stored, how it is matched, and what is retrieved.

Under this constraint, kernelized (linear) attention admits a particularly simple form. Recall that linear attention replaces the softmax kernel with a positive feature map $\phi(\cdot)$, allowing the attention computation to be reordered as

$$y_t = \frac{\phi(q_t)^\top \sum_{i \le t} \phi(k_i) v_i^\top}{\phi(q_t)^\top \sum_{i \le t} \phi(k_i)}. \tag{5}$$

Substituting $q_t = k_t = v_t = z_t$ yields the recurrence

$$S_t = \sum_{i \le t} \phi(z_i) z_i^\top, \qquad y_t = \frac{\phi(z_t)^\top S_t}{\phi(z_t)^\top \sum_{i \le t} \phi(z_i)}, \tag{6}$$

where $S_t$ is a running state that aggregates outer products of the current representation with itself. Importantly, the state update can be written incrementally as

$$S_t = S_{t-1} + \phi(z_t) z_t^\top, \tag{7}$$

optionally with a decay factor $S_t = \lambda S_{t-1} + \phi(z_t) z_t^\top$ to ensure stability. No token–token interaction matrix is ever formed; all computation proceeds through a streaming state update and a local readout.

This formulation reveals a direct structural correspondence between linear attention with collapsed QKV and state-space models (SSMs). Classical discrete-time SSMs evolve a hidden state according to

$$h_t = Ah_{t-1} + Bx_t, \qquad y_t = Ch_t, \tag{8}$$

where $A$ controls state dynamics and $B$ injects input into the state. In the linear-attention recurrence above, $S_t$ plays the role of the hidden state, the outer-product term $\phi(z_t) z_t^\top$ acts as an input-dependent update, and the optional decay corresponds to a stable transition operator. The key difference is that attention employs an input-conditioned readout, $y_t = \phi(z_t)^\top S_t$, rather than a fixed observation matrix. Conceptually, linear attention therefore behaves as a state-space model with adaptive, content-dependent observation.

Collapsing Q, K, and V removes explicit content-based routing and converts attention into a dynamical memory system closely related to fast-weight models and Hebbian associative updates. The resulting model emphasizes continuous temporal integration and efficient long-range aggregation rather than selective retrieval and symbolic addressing. This unification clarifies why linear attention and modern SSMs share similar scaling properties, streaming behavior, and inductive biases, while also explaining their limitations in tasks requiring sharp, discrete information routing. From an architectural perspective, the QKV collapse highlights a continuum between programmable memory (attention) and dynamical systems (SSMs), reinforcing the view that representational structure, not scale alone, determines the qualitative behavior of sequence models.

## A.2. 2D Positional Encodings

We use 2D positional encodings in the "+" variants to restore directional asymmetry in attention when projection sharing (e.g., $Q = K$) produces symmetric attention maps ($QK^\top = KK^\top$).

**Construction:** We define a fixed 2D sinusoidal positional encoding

$$P \in \mathbb{R}^{n \times n \times m},$$

where $n$ is the sequence length and $m$ the positional embedding dimension. Each entry $P_{i,j}$ encodes the relative interaction between query position $i$ and key position $j$, allowing the model to distinguish directional relationships ($i < j$ vs. $i > j$).

**Integration into Attention:** Given raw attention scores

$$A = QK^\top \in \mathbb{R}^{n \times n},$$

we broadcast $A$ along a channel dimension, add the positional encoding

$$A' = A + P,$$

and apply a $1 \times 1$ convolution (linear projection) to map $A' \in \mathbb{R}^{n \times n \times m} \to \mathbb{R}^{n \times n}$.

**Intuition:** This modifies attention to combine content-based similarity with positional/directional bias, breaking symmetry caused by projection sharing and enabling order-sensitive behavior.

### A.3. Additional Synthetic and Vision Results

#### A.3.1. SYNTHETIC RESULTS

Figure 2 shows the loss over time for the synthetics tasks. Figure 3 displays sample attention maps. It should be noted that the attention maps of the KV (Q=K≠V) transformer exhibit symmetry around the line $y = x$. Notable patterns can be observed within the attention maps. For instance, in the reversing task, the QKV model has learned to take care of the token located at the flipped index of itself. However, it also allocates some attention to values near the flipped index. This behavior arises because the model does not require precise, strict attention to solve this problem, but rather benefits from an approximate, noisy attention map. Figure 4 shows the code to compute and normalize the self attention map, plus visualization of maps.

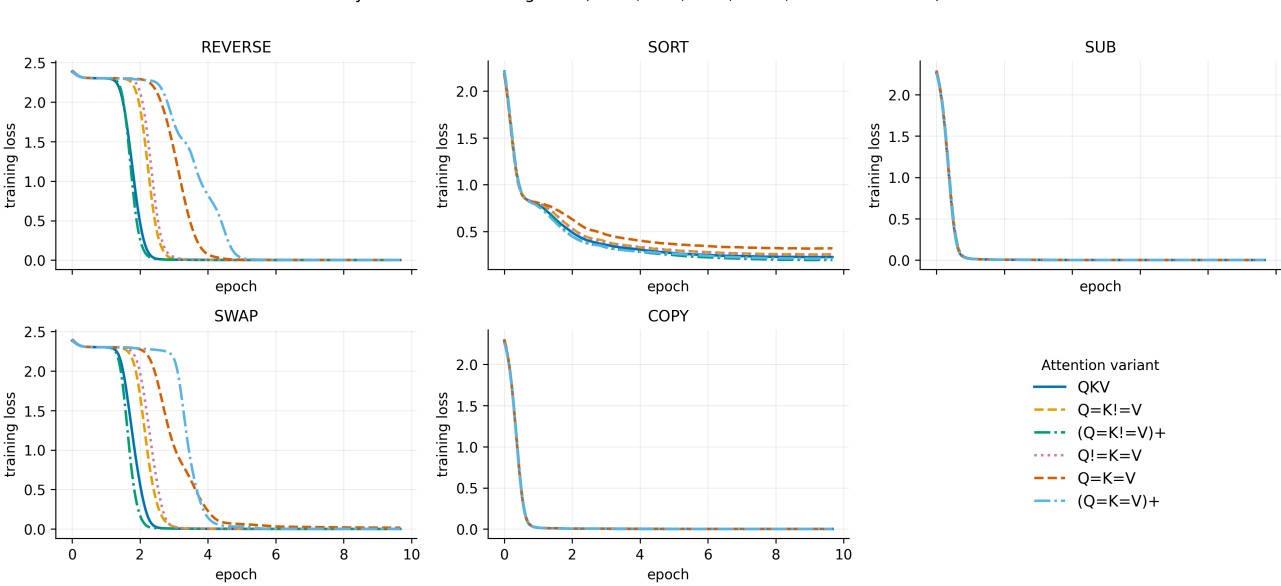

*Figure 2.* Training loss on the five synthetic tasks for all six attention variants (mean of three seeds; representative configuration $d$=64, $L$=2, $H$=4, sequence length 16, ten epochs). The position-wise tasks SUB and COPY are solved almost immediately by every variant, whereas the directional tasks REVERSE and SWAP reveal a clear convergence ordering: QKV and (Q=K≠V)$^+$ converge fastest, followed by the two-projection variants Q=K≠V and Q≠K=V, while the single-projection Q=K=V converges slowest. On SORT, Q=K=V also plateaus at the highest loss. All variants eventually reach near-zero training loss on the easier tasks.

#### A.3.2. SET ANOMALY DETECTION

We apply transformers to sets (*i.e.* unordered inputs). A model is trained to find the odd one out in a set of ten images, using CIFAR-100. Nine images are drawn from a single class, and one (the anomaly) from a different class. Two sample sets are shown in Figure 5. CIFAR-100 has 60K 32×32 images over 100 classes (600 per class).

To extract high-level, low-dimensional features from the images, we employ a ResNet34 model (He et al., 2016) pretrained on ImageNet (Deng et al., 2009), discarding its classification head to obtain a 512-dimensional feature vector per image. These features are precomputed once and kept frozen; the attention model operates directly on them. Training sets are drawn from the CIFAR-100 training split and evaluation sets from the test split.

Each training example is a set of ten feature vectors—nine sampled from a randomly chosen class and one anomaly sampled from a different class, with the anomaly placed at a random position—and the label is the anomaly's index. We perform set-level classification by assigning one logit per image and applying a softmax across the ten images, yielding permutation-equivariant predictions that identify the anomalous image regardless of input order. Accordingly, the set encoder uses no positional embedding.

We vary the embedding dimension over $\{64, 256\}$ and the number of heads over $\{2, 4\}$, using two layers, the Adam optimizer with a learning rate of $10^{-3}$, and gradient clipping at 5. Each configuration is trained for 20 epochs on 10,000 sampled sets and evaluated on 2,000 held-out sets; results are averaged across configurations (see Table 3). All six attention

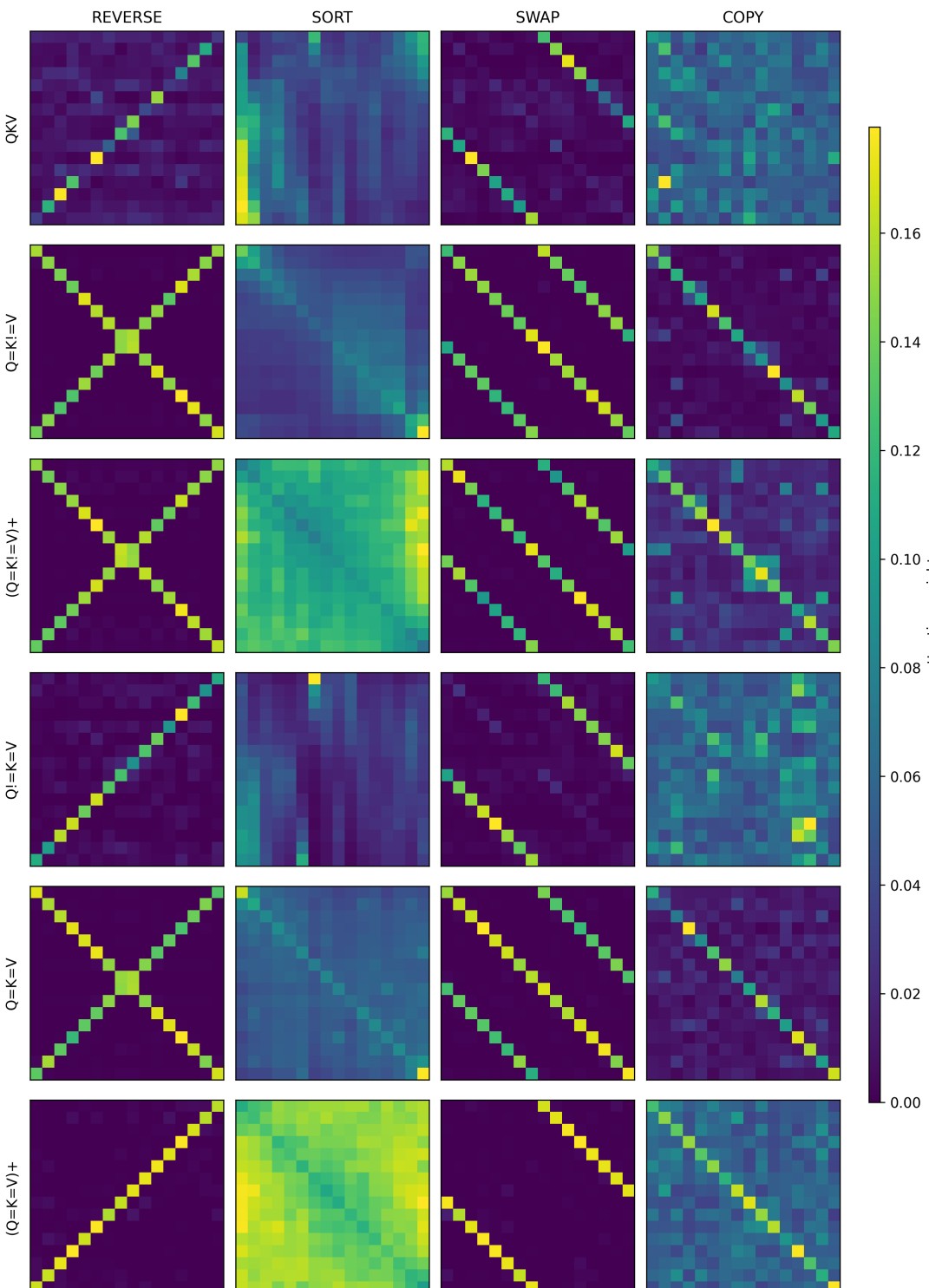

*Figure 3.* Sample last-layer self-attention maps (softmax weights averaged over heads) for one fixed input per task, across all six variants ($d$=64, $L$=2, $H$=4, sequence length 16). Each task induces an interpretable pattern: a diagonal for COPY (attend to self), an anti-diagonal for REVERSE (position $i$ attends to $n-1-i$), and shifted off-diagonal blocks for SWAP. The symmetric-projection variants (Q=K$\neq$V, Q=K=V) cannot represent a purely directional map and instead produce symmetric "X"-shaped patterns on REVERSE/SWAP; adding the 2D positional injection, $(X)^+$, breaks this symmetry—e.g. $(Q=K=V)^+$ recovers a clean anti-diagonal on REVERSE. QKV and Q$\neq$K=V, whose maps are asymmetric by construction, show the directional pattern directly.

```python
def normalize_kkt_diagonal(K):
    """Normalizes the diagonal of K @ K^T."""
    S = K @ K.T  # Calculate K @ K^T
    diagonal = torch.diag(S)
    scaling_factor = torch.mean(diagonal)  # Or torch.linalg.norm(diagonal) or other scaling factor.
    normalized_S = S.clone()

    # Iterate and fill each diagonal element
    for i in range(normalized_S.shape[0]):
        normalized_S[i, i] = diagonal[i] / scaling_factor

    return normalized_S, S

# Example Usage:
n = 10
h = 30
K = torch.randn(n, h)  # Generate a random K matrix
S_normalized, S = normalize_kkt_diagonal(K)
print("Normalized S:\n", S_normalized)
```

*Figure 4.* Top) Code to compute and normalize the self attention map. Bottom) un-normalized and normalized (right) attention maps.

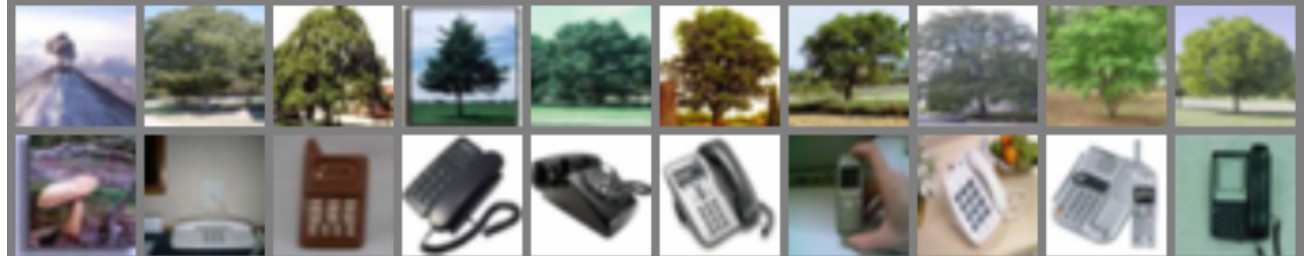

*Figure 5.* Two sets of samples from the anomaly detection dataset, with the first image in each set representing the anomaly.

variants perform at or above the QKV baseline, and the positional-injection variants perform best: $(Q=K\neq V)^+$ attains the highest odd-one-out accuracy, followed closely by $(Q=K=V)^+$.

### A.3.3. IMAGE SEGMENTATION

Hwa et al. (2025) did some experiments based on an earlier version of our work (Borji, 2023). They applied the proposed models to a more complex and larger-scale scenario. The task was semantic segmentation of abdominal MRI slices by labeling each pixel to belong to one of three categories: large bowel, small bowel, or stomach.

They implemented several models with QKV (default) and KV (corresponding to Q=K≠V here) attention variants, as detailed below. They skip the K variant (corresponding to Q=K=V here) to allocate computational resources on the more competitive variants. All models share a convolutional decoder adapted from SETR (Fig. 6, left side). They modified the decoder to halve the feature dimensions during upsampling, reducing the overall parameter count (Fig. 6, right side).

They implemented the SETR encoder as outlined in (Zheng et al., 2021), using a ViT-B/16 backbone with the feature dimension $D = 768$, number of heads $H = 12$, and number of layers $L = 12$ with both QKV and KV attention mechanisms. They refer to these architectures as SETR-QKV and SETR-KV, respectively.

Furthermore, they explored SETR-KV+Pos, where they introduced positional encoding within the KV attention block to create asymmetry. The 2D positional encoding dimension $m$ was set to 50. Additionally, they constructed two models with a hybrid encoder. Drawing inspiration from TransUNet (Chen et al., 2021), they integrated the first four convolutional layers of the ResNet-50 architecture (He et al., 2016) into encoder to capture higher-dimensional features before the patch embedding stage. In the fourth layer, they increased the number of blocks from 6 to 9 to improve feature extraction while maintaining a feature dimension of 1024. Unlike the approach in (Chen et al., 2021), no skip connections were used. They refer to these models as SETR-QKV-CE and SETR-KV-CE, respectively.

Finally, they developed an additional hybrid model using a Convolutional Vision Transformer (CvT) (Wu et al., 2021) as the encoder. The models SETR-QKV-CVT and SETR-KV-CVT utilize a CvT-13 encoder, with the multi-head attention (MHA) in the Convolutional Transformer Blocks implemented with QKV and KV attention, respectively.

All models were trained for 100 epochs without early stopping to ensure comparable results. The input resolution was set at $224 \times 224$ and a fixed patch size of $16 \times 16$ was chosen. The AdamW optimizer with a learning rate of 1e-4 and polynomial learning rate scheduling with factor 0.9 were used. Furthermore, a batch size of 32 was chosen for training. During training, on-the-fly data augmentation was applied, namely horizontal flipping, vertical flipping, shift scale rotate, coarse dropout, and random bright contrast, each having 50% probability of being applied. All models were trained from scratch (*i.e.* no use of pretrained backbones).

The medical image dataset used was UW-Madison GI Tract Image Segmentation (happyharrycn et al., 2022) which consists of abdominal MRI slices. Annotations of the three classes were provided in the form of run-length encoded organ segmentations. During preprocessing, they transformed the RLE ground truth data into 2D grayscale multi-class masks. The dataset was split into training, validation, and test sets with a ratio of 80 : 16 : 4.

The performance metrics computed for the tested architectures include the Jaccard index and the weighted Jaccard index (Table 12). Model complexity is represented by the number of learnable parameters, while computational efficiency is assessed by the number of multiply-accumulate operations (MACs) (collected through the `torchinfo` and `ptflops` python modules). Their results indicate that all tested attention variants perform comparably well or slightly better than

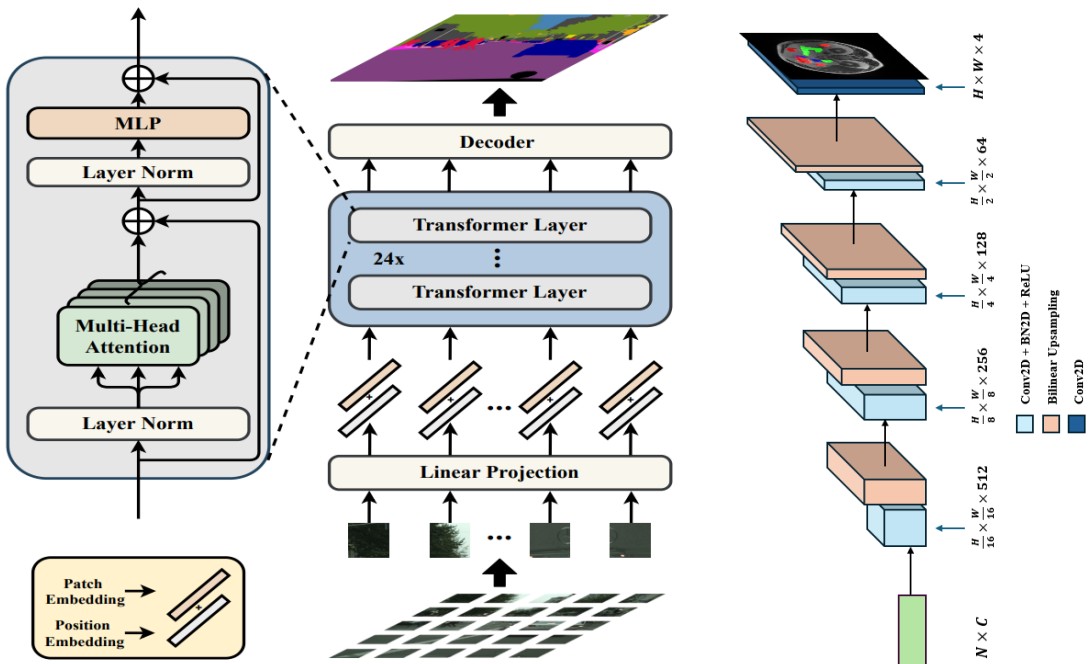

*Figure 6.* Left: The standard SETR architecture (Zheng et al., 2021). Right: The SETR-PUP decoder. It is modified to also reduce feature dimensions during upsampling.

*Table 12.* The results of semantic segmentation experiments. No performance drop was observed among most of the KV variants, while simultaneously seeing a reduction in parameter count and MACs. The asterisk (*) indicates that the MACs calculation does not account for the calculations related to 2D positional encoding.

|  | Jaccard | Weighted Jaccard | # Parameters (M) | MACs (G) |
|---|---|---|---|---|
| SETR-QKV | 0.8718 | 0.8653 | 91.04 | 21.07 |
| SETR-KV | 0.8727 | 0.8667 | 83.96 | 19.68 |
| SETR-KV+Pos | 0.8750 | 0.8691 | 83.96 | 19.97* |
| SETR-QKV-CE | 0.8990 | 0.8946 | 103.14 | 25.02 |
| SETR-KV-CE | **0.9015** | **0.8960** | 96.05 | 23.68 |
| SETR-QKV-CVT | 0.8846 | 0.8786 | 22.9 | 9.86 |
| SETR-KV-CVT | 0.8807 | 0.8755 | **21.3** | **9.49** |

their corresponding QKV implementations, while also demonstrating a reduction in both parameter count and MACs of approximately 10%.

For model details and additional results please refer to (Hwa et al., 2025).

## A.4. Additional LLM Results

This section provides additional visualizations and detailed results for the language modeling experiments described in Section 4.3. We present comprehensive comparisons of projection sharing variants, head sharing mechanisms, and their combinations across both 300M and 1.2B parameter scales.

Figures 7 and 8 visualize the core trade-offs between model quality (perplexity) and inference efficiency (KV cache reduction). Figure 9 synthesizes these results into an efficiency-quality Pareto frontier, demonstrating that projection sharing and head sharing operate on complementary optimization axes. Figures 10 and 11 show complete training curves, confirming that quality rankings remain stable throughout training and across model scales. Table 13 provides a comprehensive reference for all evaluated variants. These visualizations reveal that Q≠K=V achieves the best balance between cache reduction and model quality, while combined approaches like Q-MQA push the efficiency frontier to near-theoretical limits with 96.9% cache reduction (at 300M scale). The consistency of results across scales validates the reliability of our architectural comparisons and provides confidence in the generalizability of these findings to larger production models.

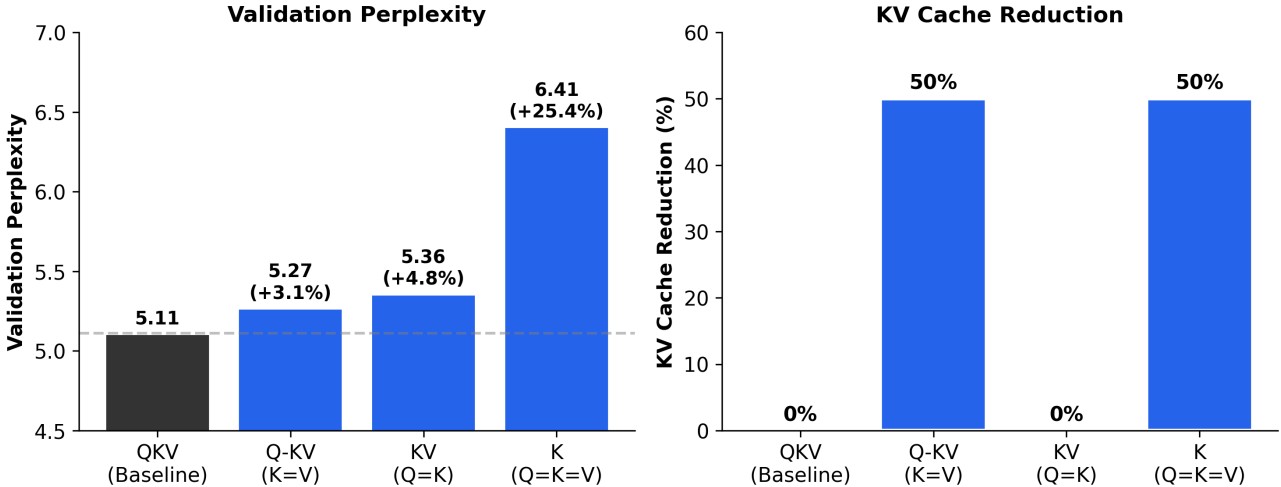

*Figure 7.* **Projection sharing variants on 300M parameter LLMs trained on 10B tokens.** Left: Validation perplexity (lower is better). Right: KV cache reduction (higher is better). Q≠K=V achieves 50% cache reduction with only 3.1% perplexity degradation. KV (Q=K≠V) provides no cache benefit despite 4.8% degradation due to still requiring separate K and V caches. K (Q=K=V) causes catastrophic 25.4% degradation, making it impractical.

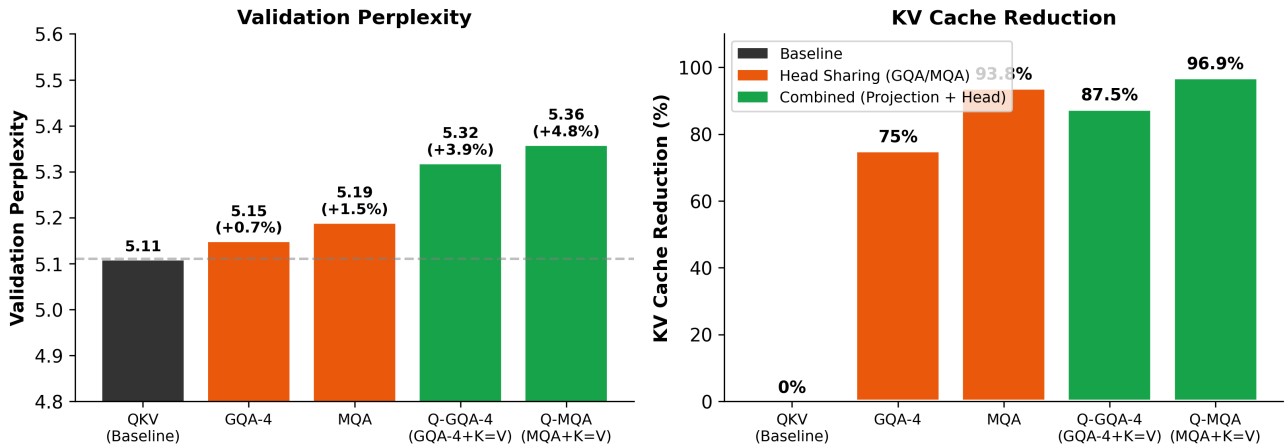

*Figure 8.* **Head sharing and combined approaches on 300M parameter LLMs.** Left: Validation perplexity. Right: KV cache reduction. Orange bars: head sharing only (GQA-4, MQA). Green bars: combined projection + head sharing (Q-GQA-4, Q-MQA). Combined approaches achieve up to 96.9% cache reduction while maintaining less than 5% perplexity degradation, demonstrating that projection sharing and head sharing are complementary optimization axes.

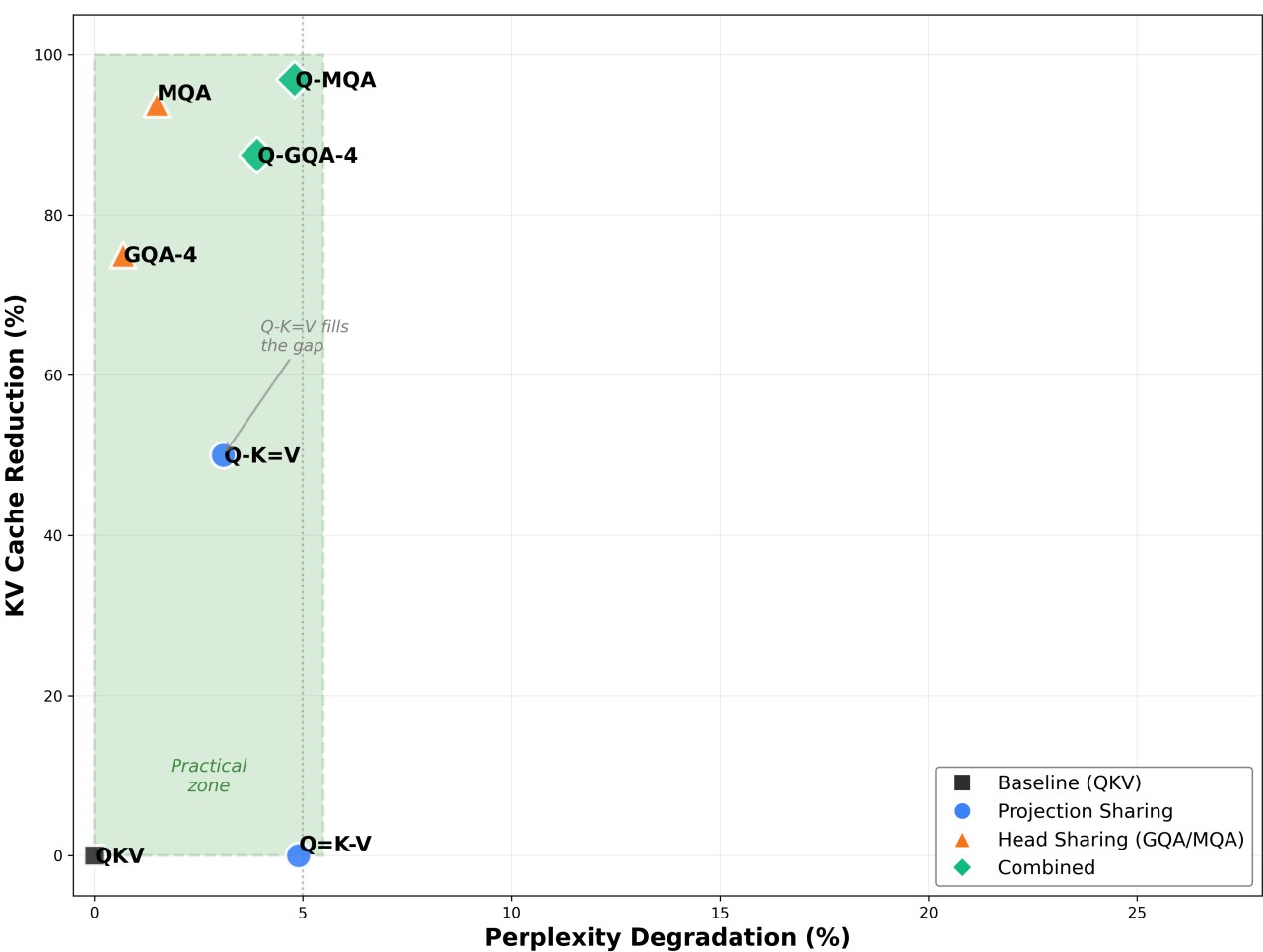

*Figure 9.* **Efficiency-quality Pareto frontier for attention variants.** Projection sharing (blue circles) and head sharing (orange triangles) occupy complementary regions. Combined approaches (green diamonds) achieve the highest cache reductions. The shaded region indicates practical deployment zone (<5% perplexity degradation). Q≠K=V fills the gap between QKV baseline and head-sharing methods, providing 50% cache reduction with only 3.1% degradation.

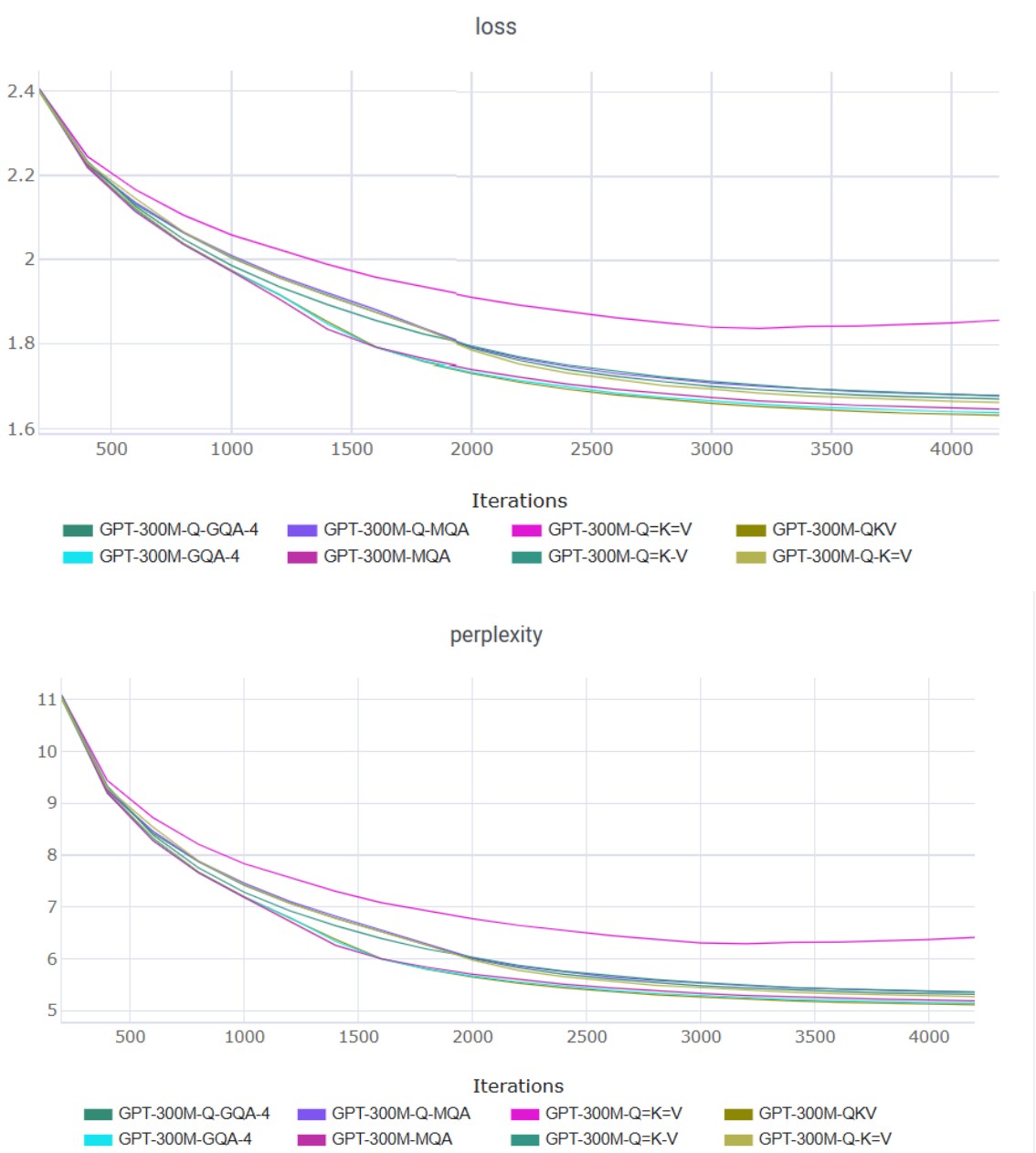

*Figure 10.* **Validation curves for 300M parameter models.** Top: Validation loss. Bottom: Validation perplexity over 10B training tokens. Q≠K=V (dark teal) matches baseline QKV (olive) closely on held-out data, achieving 50% cache reduction with only 3.1% perplexity degradation. Q=K≠V (light pink) shows higher validation loss, confirming suboptimal generalization. All head-sharing and combined variants converge to practical validation performance.

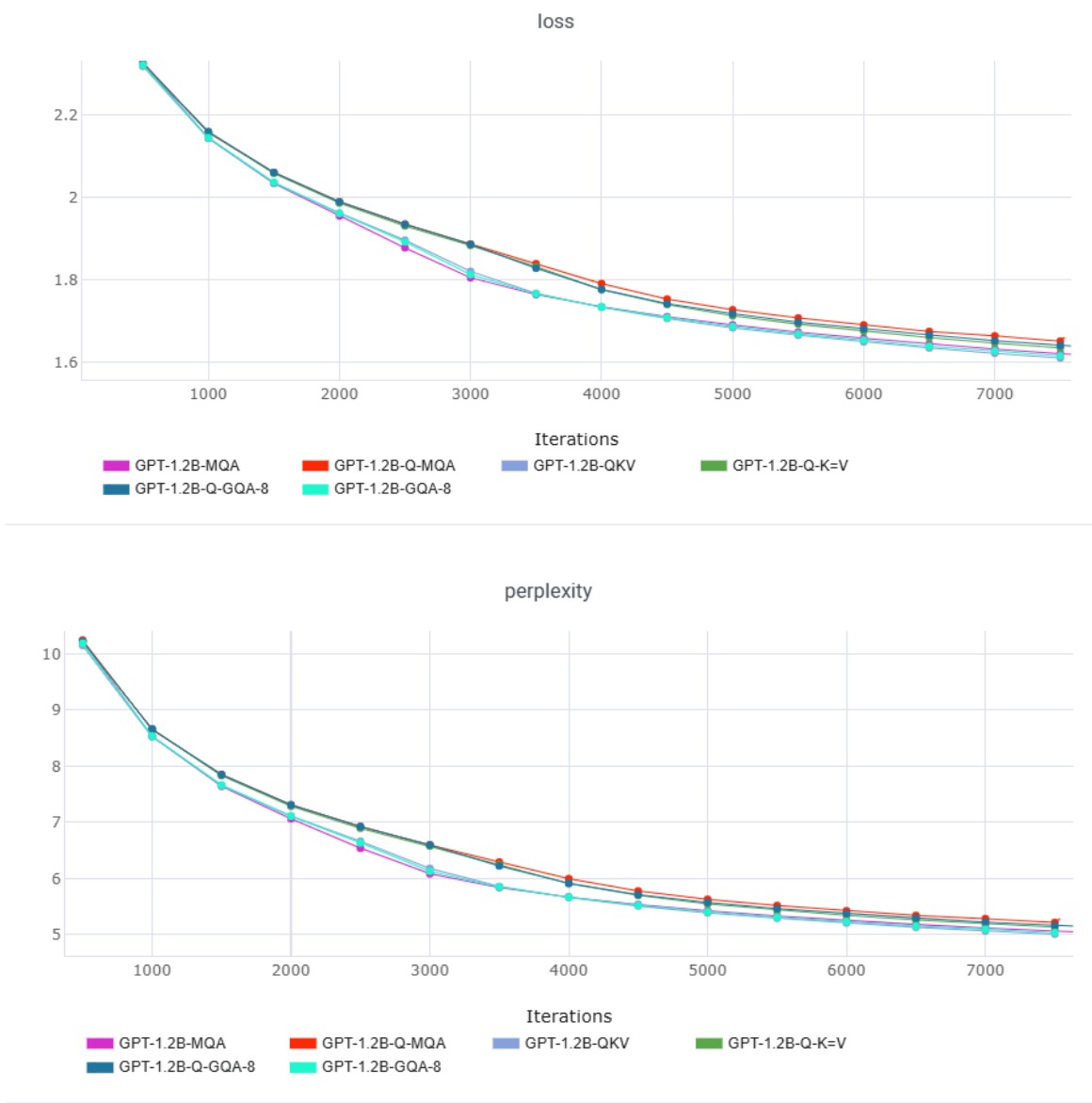

*Figure 11.* **Validation curves for 1.2B parameter models.** Top: Validation loss. Bottom: Validation perplexity over 10B training tokens. Rankings on held-out data remain consistent with 300M scale. Q≠K=V (green) and head-sharing variants track baseline QKV (gray/brown) closely, while combined approaches (Q-GQA-8, Q-MQA) maintain < 5% degradation with 88-98.5% cache reduction, confirming scalability of our findings.

*Table 13.* Comprehensive summary of all attention mechanism variants evaluated. PE = Positional Encoding. Cache column shows what must be stored during autoregressive generation. Cache reduction and perplexity degradation were reported for 300M parameter models. The "—" entries in the PPL Δ column correspond to $(X)^+$ variants, which were evaluated only on non-causal tasks (vision and synthetic); see Section 2, "Scope of $(X)^+$ variants."

| # | Notation | Projections | Cache | Cache↓ | PPL Δ | Key Insight |
|---|---|---|---|---|---|---|
| **Baseline** | | | | | | |
| 1 | QKV | Q, K, V | K+V | 0% | 0% | Standard attention |
| **Projection Sharing** | | | | | | |
| 2 | Q=K≠V | Q=K, V | K+V | 0% | +4.9% | Symmetric, no cache benefit |
| 3 | $(Q=K \neq V)^+$ | Q=K, V, +PE | K+V | 0% | — | Adds 2D PE for asymmetry |
| 4 | Q≠K=V | Q, K=V | K | 50% | +3.1% | **50% Cache reduction (Optimal)** |
| 5 | Q=K=V | Q=K=V | K | 50% | +25.4% | Too constrained |
| 6 | $(Q=K=V)^+$ | Q=K=V, +PE | K | 50% | — | PE partially recovers quality on synthetic |
| **Head Sharing (Comparison Baselines)** | | | | | | |
| 7 | GQA-4 | Q, K, V (4 groups) | K+V | 75% | +0.7% | 4 groups, 16 heads total |
| 8 | MQA | Q, K, V (1 head) | K+V | 93.8% | +1.5% | Single KV head for all Q |
| **Combined: Projection + Head Sharing** | | | | | | |
| 9 | Q-GQA-4 | Q, K=V (4 groups) | K | 87.5% | +3.9% | K=V within each group |
| 10 | Q-MQA | Q, K=V (1 head) | K | 96.9% | +4.8% | K=V on single head |

KEY TAKEAWAYS FROM ADDITIONAL RESULTS

The visualizations and comprehensive comparisons in this appendix support several important conclusions:

1. **Q≠K=V is the clear winner for projection sharing.** It achieves 50% cache reduction with only 3.1% perplexity degradation at 300M scale and 2.48% at 1.2B scale, representing a new point on the efficiency-quality Pareto frontier.

2. **Cache reduction, not parameter reduction, drives practical benefits.** While all projection sharing variants reduce parameters, only K=V constraints reduce inference memory. This explains why Q=K≠V fails to provide deployment advantages despite competitive training quality.

3. **Projection and head sharing are strictly complementary.** Combined approaches achieve 87.5% (Q-GQA-4) to 96.9% (Q-MQA) cache reduction, enabling practical on-device inference for billion-parameter models.

4. **Quality rankings remain stable across scales.** The relative performance of all variants is consistent from 300M to 1.2B parameters, with larger models showing slightly better robustness to projection constraints.

5. **No training instabilities observed.** All variants converge smoothly without requiring specialized initialization, learning rate schedules, or architectural modifications beyond the attention mechanism itself.

These results establish projection sharing as a practical optimization for memory-efficient transformer deployment, particularly for applications requiring long contexts or high throughput in resource-constrained environments.

**Inference Wall-Clock Benchmarks.** To validate that the theoretical KV cache reductions translate to measurable deployment gains, we benchmarked all 1.2B variants on a single NVIDIA A100 GPU using bfloat16 with standard causal attention. We report both a forward-pass benchmark across batch sizes $\{1, 4, 16\}$ and sequence lengths $\{1024, 2048\}$ (Table 14), and an autoregressive generation benchmark with a 128-token prompt generating 128 new tokens (Tables 15a and 15b). All variants share identical hardware, software, and runtime configuration.

Across all configurations, Q≠K=V achieves 6.5–6.9% peak memory reduction, 4.4–5.3% higher decode throughput, and 4.3–5.0% lower per-token latency relative to QKV. The 6.5–6.9% total memory reduction reflects KV cache as one component of peak memory; activations, weights, and workspace dominate the remainder. The structural 50% KV cache reduction is fully realized in production serving systems (e.g., vLLM) where K and V are allocated separately per decode step. Combined approaches push further: Q-MQA achieves 12.8–13.6% memory reduction and 11.7–13.2% throughput improvement, approaching the cache-bound limit for transformer generation.

*Table 14.* Forward-pass inference benchmark on a single A100 (1.2B models, bf16). All variants reduce peak memory and improve throughput versus the QKV baseline at every batch size and sequence length tested.

| Model | Batch | SeqLen | Mem (GB) | Tok/s | Latency (ms) |
|---|---|---|---|---|---|
| QKV | 1 | 1024 | 5.551 | 5474 | 187.1 |
| | 4 | 1024 | 6.482 | 5882 | 696.3 |
| | 16 | 1024 | 10.206 | 6139 | 2668.7 |
| | 1 | 2048 | 6.401 | 4849 | 422.3 |
| | 4 | 2048 | 9.874 | 4996 | 1639.6 |
| | 16 | 2048 | 23.766 | 4988 | 6569.0 |
| Q≠K=V | 1 | 1024 | 5.173 | 5799 | 176.6 |
| | 4 | 1024 | 6.079 | 6201 | 660.6 |
| | 16 | 1024 | 9.703 | 6522 | 2512.1 |
| | 1 | 2048 | 6.015 | 5088 | 402.5 |
| | 4 | 2048 | 9.437 | 5249 | 1560.6 |
| | 16 | 2048 | 23.128 | 5235 | 6259.2 |
| GQA-8 | 1 | 1024 | 5.019 | 5945 | 172.3 |
| | 4 | 1024 | 5.944 | 6383 | 640.6 |
| | 16 | 1024 | 9.717 | 6717 | 2430.3 |
| | 1 | 2048 | 5.857 | 5203 | 393.6 |
| | 4 | 2048 | 9.351 | 5367 | 1526.4 |
| | 16 | 2048 | 23.340 | 5369 | 6111.3 |
| MQA | 1 | 1024 | 4.834 | 6126 | 167.3 |
| | 4 | 1024 | 5.768 | 6574 | 623.1 |
| | 16 | 1024 | 9.499 | 6900 | 2374.4 |
| | 1 | 2048 | 5.687 | 5325 | 384.6 |
| | 4 | 2048 | 9.163 | 5486 | 1493.3 |
| | 16 | 2048 | 23.067 | 5486 | 5973.2 |
| Q-GQA-8 | 1 | 1024 | 4.904 | 6041 | 169.5 |
| | 4 | 1024 | 5.835 | 6484 | 631.7 |
| | 16 | 1024 | 9.560 | 6801 | 2408.9 |
| | 1 | 2048 | 5.755 | 5261 | 389.3 |
| | 4 | 2048 | 9.228 | 5431 | 1508.3 |
| | 16 | 2048 | 23.119 | 5431 | 6033.9 |
| Q-MQA | 1 | 1024 | 4.815 | 6138 | 166.8 |
| | 4 | 1024 | 5.722 | 6596 | 621.0 |
| | 16 | 1024 | 9.349 | 6927 | 2365.2 |
| | 1 | 2048 | 5.658 | 5338 | 383.7 |
| | 4 | 2048 | 9.082 | 5490 | 1492.3 |
| | 16 | 2048 | 22.779 | 5491 | 5967.7 |

**Perplexity Across Context Lengths.** To confirm that projection sharing's quality cost does not compound with longer contexts, we evaluated all 1.2B variants at three sequence lengths (512, 1024, 2048) on a held-out SlimPajama validation subset. Table 16 reports relative perplexity degradation versus QKV at each length. These results use fixed-length truncation without document-packed inputs; absolute perplexities are therefore not directly comparable to Table 9, and short-context values may be inflated by low-context positions. We include them to characterize relative rankings across lengths rather than as precise degradation estimates.

The relative rankings are stable across all sequence lengths, confirming that the efficiency hierarchy in Table 9 generalizes across context lengths. Q≠K=V's degradation decreases from 5.4% at 512 tokens to 2.2% at 2048 tokens, aligning closely with its +2.48% in Table 9. MQA shows a slight apparent advantage over QKV under this evaluation; we note this does not fully align with Table 9 (+1.06% degradation there), and attribute the discrepancy to the truncation-based evaluation methodology. Q-MQA results were unstable on this evaluation subset and are omitted.

*Table 15.* Autoregressive generation benchmark on a single A100 (1.2B models, bf16, 128-token prompt, 128 tokens generated). (Left) raw measurements. (Right) savings versus QKV. Q≠K=V consistently outperforms QKV across all configurations.

| Model | Batch | Mem (GB) | Tok/s | ms/tok |
|-------|-------|----------|-------|--------|
| QKV | 1 | 5.347 | 25.90 | 38.61 |
| | 4 | 5.667 | 32.20 | 124.21 |
| Q≠K=V | 1 | 4.976 | 27.05 | 36.96 |
| | 4 | 5.296 | 33.90 | 118.00 |
| GQA-8 | 1 | 4.793 | 27.80 | 35.97 |
| | 4 | 5.113 | 35.08 | 114.03 |
| MQA | 1 | 4.630 | 29.02 | 34.46 |
| | 4 | 4.951 | 36.34 | 110.07 |
| Q-GQA-8 | 1 | 4.700 | 28.29 | 35.35 |
| | 4 | 5.020 | 35.80 | 111.74 |
| Q-MQA | 1 | 4.619 | 28.93 | 34.56 |
| | 4 | 4.939 | 36.45 | 109.73 |

*(a)* Raw measurements.

| Model | Batch | Mem ↓ | Tok/s ↑ | ms/tok ↓ |
|-------|-------|-------|---------|----------|
| Q≠K=V | 1 | 6.9% | 4.4% | 4.3% |
| | 4 | 6.5% | 5.3% | 5.0% |
| GQA-8 | 1 | 10.4% | 7.3% | 6.8% |
| | 4 | 9.8% | 8.9% | 8.2% |
| MQA | 1 | 13.4% | 12.0% | 10.7% |
| | 4 | 12.6% | 12.9% | 11.4% |
| Q-GQA-8 | 1 | 12.1% | 9.2% | 8.4% |
| | 4 | 11.4% | 11.2% | 10.0% |
| Q-MQA | 1 | 13.6% | 11.7% | 10.5% |
| | 4 | 12.8% | 13.2% | 11.7% |

*(b)* Savings versus QKV.

*Table 16.* Relative perplexity degradation (%) versus QKV at varying sequence lengths for 1.2B models. Relative rankings are stable across context lengths. Under this evaluation, degradation decreases with sequence length for all variants, suggesting the quality-efficiency trade-off does not worsen in the long-context regime where cache savings matter most. Results use fixed-length truncation; see text for methodology caveats.

| Model | 512 | 1024 | 2048 |
|-------|-----|------|------|
| Q≠K=V | +5.4% | +3.7% | +2.2% |
| GQA-8 | +0.8% | +0.6% | +0.3% |
| MQA | −2.8% | −1.9% | −1.1% |
| Q-GQA-8 | +1.2% | +0.8% | +0.5% |

## A.5. Full Training Configuration

We provide complete training and architectural details for the language modeling experiments described in Section 4.3, extending the summary in Section 3.3.

**Architecture.** The 300M models use 20 transformer layers, embedding dimension $d = 1024$, 16 attention heads (head dimension 64), and feed-forward dimension 4096. The 1.2B models use 22 layers, $d = 2048$, 32 attention heads (head dimension 64), and feed-forward dimension 8192. Both configurations use GELU activation in the feed-forward sublayers. Pre-Norm LayerNorm ($\epsilon = 10^{-5}$) is applied before each attention and feed-forward sublayer. Input and output embeddings are tied, with vocabulary size 50,304 using the GPT-2 tokenizer. Positional information is encoded via learned absolute position embeddings with maximum sequence length 2048. Residual dropout is set to 0.1.

**Optimization.** All models are trained from scratch with AdamW ($\beta_1 = 0.9$, $\beta_2 = 0.95$, weight decay 0.1, gradient clipping at norm 1.0). The learning rate schedule is 1000-step linear warmup to a peak of $6 \times 10^{-5}$, followed by cosine decay to a minimum of $6 \times 10^{-6}$.

**Infrastructure.** Training uses bfloat16 mixed precision on $8 \times$ NVIDIA A100 40GB GPUs with distributed data parallelism and gradient accumulation of 36 steps. The 300M models are trained for 4,238 steps ($\sim$10B tokens); the 1.2B models for 8,475 steps ($\sim$10B tokens). Validation perplexity is evaluated every 500 steps on a held-out 10M-token subset of SlimPajama. The only architectural difference across variants is the attention projection mechanism; all other components are held identical to ensure a controlled comparison.

## A.6. Summary of Projection- and Head-Sharing Variants

Table 17 summarizes the practical trade-offs among the proposed attention variants. The results reveal a clear hierarchy. Among the projection-sharing methods, Q≠K=V provides the most favorable balance between model quality and efficiency, preserving asymmetric attention while reducing the KV cache by 50%, making it a practical replacement for standard QKV attention. In contrast, Q=K≠V performs competitively on non-causal tasks such as vision, but offers no inference memory advantage because both keys and values must still be cached separately. The fully shared Q=K=V variant represents the most aggressive simplification and substantially reduces model capacity, making it unsuitable for language modeling despite its simplicity. Finally, projection sharing is complementary to head sharing: combining Q≠K=V with GQA or MQA yields additional cache reductions while maintaining competitive downstream performance, providing attractive operating points for memory-constrained deployment.

*Table 17.* Summary of projection-sharing variants.

| Variant | Proj. | Attention | KV Cache | LLM Quality | Main Advantage | Recommended Use |
|---------|-------|-----------|----------|-------------|----------------|-----------------|
| QKV | 3 | Asymmetric | Baseline | Excellent | Maximum flexibility | General purpose |
| Q≠K=V | 2 | Asymmetric | 50%↓ | Excellent | Best quality–efficiency trade-off | **Default replacement for QKV** |
| Q=K≠V | 2 | Symmetric | Baseline | Moderate | Works well for non-causal tasks | Vision, sets |
| Q=K=V | 1 | Symmetric | 50%↓ | Poor | Maximum simplification | Not recommended for LLMs |
| Q-GQA | 2 | Asymmetric | 87.5%↓ | Very good | Combines projection and head sharing | Edge deployment |
| Q-MQA | 2 | Asymmetric | 96.9%↓ | Good | Maximum cache compression | Resource-limited devices |

