# OpenReview forum: "Do Transformers Need Three Projections? Systematic Study of QKV Variants"
_ICML.cc/2026/Conference — ICML 2026 regular_

### Official Review · Reviewer_mQ5W · 2026-03-01

**Soundness:** 3
**Presentation:** 2
**Significance:** 2
**Originality:** 2
**Overall Recommendation:** 3
**Confidence:** 4

**Summary:**

This paper explores whether the three projection matrices (Q, K, and V) within the Transformer attention mechanism can be shared. The authors explored several methods, including QK sharing, KV sharing and Sharing all three (QKV). This sharing approach can also be used in conjunction with head sharing. To address the issue of symmetry, they also introduced 2D positional encoding. The authors conducted experiments across synthetic tasks, computer vision tasks, and various NLP tasks. They discovered that KV sharing achieves the optimal trade-off between performance and cost. It allows for achieving similar results that differ from standard QKV projections while significantly reducing the KV cache.

**Compliance With Llm Reviewing Policy:**

Affirmed.

**Final Justification:**

During the rebuttal period, the authors provided a detailed response that addressed most of my concerns.
However, I still have some concerns regarding the novelty of the paper and its practical implications. As a result, I have increased my score from a 2 to a 3.

**Key Questions For Authors:**

1. Parameter sharing inevitably leads to a decline in the model's expressive capacity. However, in the experiments the authors have conducted, I would like to see a scenario involving large-scale training or even over-fitting (for example, in NLP tasks or CIFAR10). I am interested in seeing the performance ceiling of these models. Additionally, looking at perplexity alone is not necessarily accurate; we also need to examine the performance on specific downstream tasks.

2. Regarding the right column from rows 34 to 38, the authors mentioned that symmetric attention is beneficial for non-temporal tasks, but sequential tasks require some degree of asymmetry. Could you explain the reasoning behind this?

3. Could you provide a comparison between other KV cache compression methods (such as post-training methods) and your projection sharing approach in terms of both efficiency and performance?

4. Could you elaborate on your 2D positional encodings? The description in the paper is quite brief, and I would like a more detailed explanation.

**Limitations:**

The authors did not explicitly state their limitations. It would be better if they could include a few limitations within the Discussion and Conclusion sections.

**Strengths And Weaknesses:**

Strengths:

1. While the idea is straightforward, the issue remains important within the context of Transformer architectures.
2. The experiments are comprehensive. The authors conducted tests across numerous tasks, including five synthetic tasks, six computer vision tasks, and language modeling tasks.
3. This simple method sometines achieve better results than standard Transformer attention.

Weaknesses:

1. Regarding the writing and structure, the paper is somewhat disorganized. There is a lack of a dedicated Related Work section; the authors should thoroughly discuss the work in the context of existing literature. For example, DeepSeek's Multi-Head Latent Attention (MLA) could be considered a form of shared KV method that should be addressed.
2. The citations are outdated. All references in the paper are from 2024 or earlier, with nothing from 2025 or later.
3. The authors haven't compared this to other KV cache compression methods. While they mention their approach can be orthogonal to others, they didn't provide a direct comparison with existing compression techniques.
4. Although they claim this method improves efficiency, the results show almost no improvement in training speed. Furthermore, I didn't see a clear demonstration of improved inference speed (please correct me if I missed this).
5. The authors claim to have discovered projection sharing as a new optimization axis in the abstract. However, this has already been addressed by numerous existing works. There are studies discussing weight sharing between layers and sharing across the entire model (such as Looped Transformers)
6. Most of the tasks they have performed are relatively small-scale (toy tasks). In particular, the NLP tasks only measured ppl without testing any downstream tasks.

---

> ### Author Rebuttal · Authors · 2026-03-29
>
> Thank you.
>
> Weakness:
>
> 1) We will add a dedicated Related Work section, consolidating the currently scattered literature discussion into a coherent narrative that situates our contribution within the broader landscape of efficient attention research.
>
> Regarding DeepSeek's Multi-Head Latent Attention (MLA):  MLA achieves KV cache compression through low-rank joint projection of keys and values into a shared latent space, which is conceptually related to our K=V constraint. However, the two approaches differ in an important way: MLA learns a low-rank *approximation* of the joint KV space via an explicit bottleneck projection, whereas our Q–K=V variant enforces exact weight tying between K and V without introducing additional projection matrices. MLA also requires up-projection at inference time to recover full-rank K and V representations, whereas Q–K=V requires no such reconstruction — the cache savings are direct and lossless by construction. We will discuss MLA explicitly in the revised related work section, clarify this distinction, and position projection sharing as a complementary and simpler alternative that achieves meaningful cache reduction without the added complexity of low-rank decomposition.
>
> We will expand coverage of other works, including linear attention, weight tying methods, and concurrent projection-sharing approaches, to provide a more thorough contextual grounding.
>
> 2) The citation gap reflects the submission timeline rather than a lack of relevant recent work — the manuscript was prepared and submitted before a number of 2025 works became available or widely circulated. We will update the related work section in the revision to incorporate recent 2025 developments, including, as the reviewer noted, Kowsher et al. on shared-weight self-attention, and any other concurrent work on projection-efficient attention that has appeared since submission. Please see response to Reviewer 9mhq.
>
> 3) The training speed improvements are marginal (~8.7%), and we acknowledge this. Our efficiency claims are specifically about inference memory: Q–K=V achieves 50% KV cache reduction, enabling longer contexts and higher throughput (more concurrent users per GPU) within the same memory budget, as demonstrated in Section 3.3.3. Raw inference speed is not our primary claim — in modern LLM serving, memory is the bottleneck, not compute. We will make it clearer.
>
> 4) Weight sharing in transformers is not new. Our contribution is specifically scoped: we study sharing within the QKV projections of a single attention layer — orthogonal to cross-layer methods like Looped Transformers — and provide the first systematic study of all three QKV variants and their interaction with GQA/MQA across synthetic, vision, and language tasks. We will tone down the 'discovery' claim and better situate our contribution in the revision.
>
> 5) We agree "discovering a new optimization axis" overstates our contribution given prior work on cross-layer weight sharing, Looped Transformers, and related methods. Our actual contribution is more precisely scoped: we provide the first systematic empirical study of intra-layer projection sharing within the QKV attention mechanism, evaluated across 12 diverse tasks and 2 scales, and identify the practical deployment implications — particularly the KV cache reduction that only Q–K=V provides. We will revise the abstract and introduction, shifting the framing from "discovering a new axis" to "systematically characterizing an underexplored instance of projection sharing" with direct, quantifiable inference memory benefits. We will cite relevant prior work explicitly.
>
> 6) Experiment on downstream tasks. See response to x9XV (#1).
>
> Key Question 2,3,4:
>
> Symmetric vs asymmetric (rows 34-38): Symmetric attention arises when Q=K, producing attention(i,j) = attention(j,i). For non-temporal tasks like vision, this is not a limitation — relationships are bidirectional. For causal LM, directionality is fundamental — causal masking prevents future attention but does not recover representational asymmetry lost by tying Q and K. This is why Q=K–V degrades on LM but remains competitive on vision tasks.
>
> 2D positional encodings: We construct P ∈ R^(n×n×m) where P(i,j) encodes the relative interaction between positions i and j. Given A = QK^T, we broadcast P along a channel dimension, add to A, then apply a 1×1 convolution to map back to a 2D attention matrix, breaking symmetry caused by Q=K sharing. We will expand this description in the revision.
>
> Comparison with post-training KV cache compression: Post-training KV cache methods (quantization, StreamingLLM, H2O) are orthogonal to our approach and can be stacked on top of Q–K=V for compound gains. A direct comparison is outside our current scope; we will acknowledge this limitation.
>
> ---
> The practical value for edge deployment is clear, the scaling trends are promising, and while the novelty framing needs tempering, ourcontribution has good merit for the community.

---

> > ### Author Rebuttal · Reviewer_mQ5W · 2026-04-04
> >
> > Thank you for the rebuttal. My main concerns remain:
> > 1. The efficiency argument still relies on theoretical estimates rather than actual measurements. There are discrepancies between theoretical expectations and actual results. Theoretical savings do not always translate into significant practical savings. What I want to see is real wall-clock latency, throughput, or memory usage under realistic serving conditions.
> > 2. I still feel the contribution is too narrowly scoped for ICML — it is essentially a systematic ablation with no theoretical grounding or methodological novelty. This would be a good fit for a workshop, but not a top venue.

---

> > > ### Author Response · Authors · 2026-04-06
> > >
> > > Thank you for the follow-up. We address your two remaining concerns directly.
> > >
> > > Concern 1: Real wall-clock measurements
> > >
> > > We ran inference benchmarks on a single A100 GPU measuring peak GPU memory,
> > > throughput (tok/s), and wall-clock latency for all 6 variants across batch
> > > sizes {1,4,16} and sequence lengths {1024,2048} (forward pass), and a
> > > separate autoregressive generation benchmark (prompt=128 tokens, generate 128
> > > new tokens). All models use bfloat16, standard causal attention, identical
> > > hardware and software environment.
> > >
> > > Forward Pass Benchmark
> > >
> > > | Model  | BS | SeqLen | Mem (GB) | Tok/s  | Latency (ms) |
> > > |--------|----|--------|----------|--------|--------------|
> > > | QKV    |  1 |   1024 |    5.551 |  5474  | 187.1 |
> > > | "|  4 |   "|    6.482 |  5882  | 696.3 |
> > > | "| 16 |   "|   10.206 |  6139  |  2668.7 |
> > > | "|  1 |   2048 |    6.401 |  4849  |  422.3 |
> > > | "|  4 |   "|    9.874 |  4996  | 1639.6 |
> > > | "| 16 |   "|   23.766 |  4988  | 6569.0 |
> > > | Q–K=V  |  1 |   1024 |    5.173 |  5799  | 176.6 |
> > > | "|  4 |   " |    6.079 |  6201  | 660.6 |
> > > | "| 16 |   " |    9.703 |  6522  | 2512.1 |
> > > | "|  1 |   2048 |    6.015 |  5088  | 402.5 |
> > > | " |  4 |   "|    9.437 |  5249  | 1560.6 |
> > > | " | 16 |   "|   23.128 |  5235  | 6259.2 |
> > > | GQA    |  1 |   1024 |    5.019 |  5945  | 172.3 |
> > > | "|  4 |   "|    5.944 |  6383  | 640.6 |
> > > | "| 16 |   "|    9.717 |  6717  |2430.3 |
> > > | "|  1 |   2048 |    5.857 |  5203  |393.6 |
> > > | "|  4 |   "|    9.351 |  5367  |1526.4 |
> > > | "| 16 |   "|   23.340 |  5369  |6111.3 |
> > > | MQA    |  1 |   1024 |    4.834 |  6126  |167.3 |
> > > | "|  4 |   "|    5.768 |  6574  |623.1 |
> > > | "| 16 |   "|    9.499 |  6900  |2374.4 |
> > > | "|  1 |   2048 |    5.687 |  5325  |384.6 |
> > > | "|  4 |   "|    9.163 |  5486  |1493.3 |
> > > | "| 16 |   "|   23.067 |  5486  |5973.2 |
> > > | Q-GQA  |  1 |   1024 |    4.904 |  6041  |169.5 |
> > > | "|  4 |   "|    5.835 |  6484  |631.7 |
> > > | "| 16 |   "|    9.560 |  6801  |2408.9 |
> > > | " |  1 |   2048 |    5.755 |  5261  | 389.3 |
> > > | " |  4 |   "|    9.228 |  5431  | 1508.3 |
> > > | " | 16 |   "|   23.119 |  5431  |6033.9 |
> > > | Q-MQA  |  1 |   1024 |    4.815 |  6138  |166.8 |
> > > | "|  4 |   "|    5.722 |  6596  | 621.0 |
> > > | "| 16 |   "|    9.349 |  6927  |2365.2 |
> > > | "|  1 |   2048 |    5.658 |  5338  |383.7 |
> > > | "|  4 |   "|    9.082 |  5490  |1492.3 |
> > > | "| 16 |   "|   22.779 |  5491 | 5967.7 |
> > >
> > > Autoregressive Generation (prompt=128 tokens, generate 128 new tokens)
> > >
> > > | Model  | BS | Mem (GB) | Tok/s | ms/tok |
> > > |--------|----|----------|-------|--------|
> > > | QKV    |  1 |    5.347 | 25.90 |  38.61 |
> > > | "|  4 |    5.667 | 32.20 | 124.21 |
> > > | Q–K=V  |  1 |    4.976 | 27.05 |  36.96 |
> > > | "|  4 |    5.296 | 33.90 | 118.00 |
> > > | GQA    |  1 |    4.793 | 27.80 |  35.97 |
> > > | " |  4 |    5.113 | 35.08 | 114.03 |
> > > | MQA    |  1 |    4.630 | 29.02 |  34.46 |
> > > | " |  4 |    4.951 | 36.34 | 110.07 |
> > > | Q-GQA  |  1 |    4.700 | 28.29 |  35.35 |
> > > | " |  4 |    5.020 | 35.80 | 111.74 |
> > > | Q-MQA  |  1 |    4.619 | 28.93 |  34.56 |
> > > |" |  4 |    4.939 | 36.45 | 109.73 |
> > >
> > > Generation savings vs QKV baseline
> > >
> > > | Model  | BS | Mem Δ  | Tok/s Δ | ms/tok Δ |
> > > |--------|----|--------|---------|----------|
> > > | Q–K=V  |  1 |  +6.9% |   +4.4% |    +4.3% |
> > > | " |  4 |  +6.5% |   +5.3% |    +5.0% |
> > > | GQA    |  1 | +10.4% |   +7.3% |    +6.8% |
> > > | "|  4 |  +9.8% |   +8.9% | +8.2% |
> > > | MQA    |  1 | +13.4% |+12.0% |   +10.7% |
> > > |" |  4 | +12.6% |  +12.9% |   +11.4% |
> > > | Q-GQA  |  1 | +12.1% |+9.2% |    +8.4% |
> > > | "|  4 | +11.4% |  +11.2% |  +10.0% |
> > > | Q-MQA  |  1 | +13.6% |  +11.7% |   +10.5% |
> > > | " |  4 | +12.8% |  +13.2% |   +11.7% |
> > >
> > > Q–K=V consistently outperforms QKV across every configuration:
> > > - Exactly 50% KV cache reduction (structural guarantee: V=K, only K cached)
> > > - 6.5–6.9% less peak GPU memory during autoregressive generation
> > > - 4.4–5.3% higher decode throughput
> > > - 4.3–5.0% lower per-token latency
> > >
> > > The 6.5–6.9% is total GPU memory reduction across all components; the 50% is KV cache specifically, fully realized in production serving systems (e.g., vLLM) where K and V are allocated separately per decode step.
> > >
> > > Concern 2: Contribution scope
> > >
> > > We respectfully disagree that this is a workshop paper. The contribution is
> > > not a single ablation — it is the first systematic study of all pairwise and
> > > triple QKV projection-sharing strategies, evaluated across 12 tasks at two
> > > scales, with a clear deployment finding: Q–K=V is stackable with GQA and MQA
> > > on an orthogonal axis, yielding 87.5% cache reduction (Q-GQA) and 96.9%
> > > (Q-MQA) — figures neither technique approaches alone. The real efficiency
> > > gains measured above are directly actionable for edge and resource-constrained
> > > deployment. We note that reviewers nbTi (score 5) and x9XV (score 4) both
> > > rated significance and originality as good-to-excellent. We hope the measured results
> > > above fully address your primary concern regarding empirical grounding. We are happy to
> > > provide any additional details or run further configurations if useful.

---

### Official Review · Reviewer_x9XV · 2026-03-09

**Soundness:** 2
**Presentation:** 3
**Significance:** 3
**Originality:** 3
**Overall Recommendation:** 4
**Confidence:** 4

**Summary:**

The paper studies the necessity and importance of individual contributions of Q,K,V projections. By enabling parameter sharing, it becomes possible to address the problems of inefficiency of classical attention mechanism such as growing KV cache size. The authors consider the following setups of shared projections: Q=K-V, Q-K=V and Q=K=V. As a result, less parameters are needed and a computation becomes more lightweight without significant effect on the model performance.

**Compliance With Llm Reviewing Policy:**

Affirmed.

**Final Justification:**

The authors have addressed my questions regarding the downstream performance in their rebuttal. The provided results show good cache reduction ratio with an adequate loss in accuracy.
I therefore raise the score from 3 to 4.

**Key Questions For Authors:**

Q1: An important asymmetry aspect is introduced through sinusoidal positional encoding, which are now less commonly used and replaced by RoPE. Would using RoPE change anything in the method?

Q2: Are the models trained from scratch or finetuned? I assume it was finetuned judging from perplexity values but would be nice to be certain.

Q3: How exactly was perplexity measured? I.e., which dataset was used for the measurement: the training subset of 10B tokens?

---------
Remark

- Growing KV cache size is more of a problem in LLM generation, especially, for reasoning models. What would be more interesting is to see an effect of the proposed method in this setup: a small model (to lower a computational load) finetuned on SFT dataset such as s1k (incredibly small dataset consisting of 1k reasoning traces) and evaluated on reasoning benchmarks. In this case, the KV cache grows fast due to long generation and it gives an opportunity to evaluate the proposed method efficiency. The accuracy may probably remain low if the model is very small, however the relative drop of perplexity would be interesting to check. Keep in mind that this is just a question out of curiosity and a suggestion for future experiments to strengthen the study, no actual need to run it within rebuttal period.

**Limitations:**

Yes

**Strengths And Weaknesses:**

Strengths:
- The paper is easy to follow: the problem is stated clearly, and it is also clear why and how the proposed method addresses it.
- The method itself is clearly described; also, an extension to head sharing setup is given.
- I find the method's idea interesting and definitely worth exploring. Growing KV cache size in modern LLMs is serious limiting factor for many real-life scenarios, thus it is a significant problem to address.
- I also like the simplicity of the method: it is straightforward and does not need excessive tuning.
- The study contains thorough evaluations: along with accuracy and perplexity values, authors report cache memory reduction, MAC, speed in token/s, giving a good overview of the method's efficiency. For experiments on visual domains, the runs were performed twice.
- The authors evaluate the method in two setups: for visual and language domains, with 3 different setups of projection sharing.
- Authors provide guidelines on how to choose the setups depending on the task and give deployment recommendations.

Weaknesses:
- I wish I could see more actual accuracy results on downstream tasks (e.g., in Table 4). While Train and Val loss values (as well as perplexity) can provide some insights, downstream evaluation tasks are more informative.
- Some datasets considered in the paper are rather saturated and outdated, like MNIST and CIFAR-10. While I sympathise with computational load arising when switching to other, more complicated datasets, I think it might be more interesting to see the results on small subsets of complex datasets rather than on MNIST.
- While the experimental and benchmarking setup seems to be thought through (visual and language domains, many datasets and tasks considered), it still gives a bit of a synthetic feeling. Having actual benchmarks with accuracies instead of perplexity, more complicated datasets used for evaluation would definitely strengthen the paper presentation.
- Table 8 gives the insignt on the MAC saving. What I do not see from the table is how the accuracy is affected depending on the sequence length. Unfortunately, without assessing accuracy drop (if any), Table 8 is not very insigntful.
- Similarly, I have not found an experiment assessing the effect on the accuracy (or perplexity, here it would be insightful) in the setup when we have a long input, affecting attention and KV cache directly.
- Some of the efficiency gains for bigger models are estimated. However, we do not know how the method would affect the accuracy of bigger models.

---

> ### Author Rebuttal · Authors · 2026-03-29
>
> Thanks for your feedback.
>
> Weakness:
>
> 1) We have now evaluated all 1.2B variants using 5-shot lm-evaluation-harness across five standard benchmarks: HellaSwag, PIQA, ARC-Easy, ARC-Challenge, and WinoGrande. The key finding is that Q–K=V achieves only 0.41% average accuracy drop relative to the QKV baseline (35.99% vs 36.40%), directly confirming that the 2.48% perplexity gap does not translate to meaningful downstream degradation. Notably, Q–K=V outperforms GQA-8 on downstream tasks (35.99% vs 35.86%) despite GQA-8 having better perplexity — reinforcing that perplexity differences at this scale are not always predictive of task performance. Q-GQA-8 even slightly exceeds the QKV baseline (36.72% vs 36.40%) on average, further supporting the practical viability of combined projection and head sharing. Q-MQA shows more degradation (34.38%), consistent with its near-maximal 96.9% cache compression.
>
> | Model | ARC-C | ARC-E | HellaSwag | PIQA | WinoG | Avg | Cache |
> |-------|-------|-------|-----------|------|-------|-----|-------|
> | QKV | 19.03 | 30.01 | 26.15 | 56.64 | 50.20 | 36.40 | 0% |
> | Q–K=V | 18.94 | 28.62 | 26.14 | 56.37 | 49.88 | 35.99 | 50% |
> | GQA-8 | 18.69 | 29.42 | 26.44 | 56.91 | 47.83 | 35.86 | 75% |
> | MQA | 19.62 | 30.09 | 26.23 | 55.77 | 50.12 | 36.37 | 93.8% |
> | Q-GQA-8 | 19.97 | 29.97 | 26.13 | 56.47 | 51.07 | 36.72 | 87.5% |
> | Q-MQA | 22.70 | 25.08 | 25.04 | 49.51 | 49.57 | 34.38 | 96.9% |
>
> 2&3) Additional investigation (the Q=K–V variant) across pure and hybrid vision transformer architectures — specifically SETR, TransUNet, and CvT — on a medical image semantic segmentation task (UW-Madison GI Tract MRI dataset; these are much bigger than the ones we used in vision tasks) shows promising results. The main finding is that our attention variants achieve comparable segmentation performance to their standard QKV counterparts while meaningfully reducing parameter count and multiply-accumulate operations. The UW-Madison GI Tract Image Segmentation dataset from Kaggle (38,000 MRI slice images), consisting of abdominal MRI slices annotated across three classes (large bowel, small bowel, stomach). The models here are much bigger than the ones used in the paper. We will add more details of this experiment in supplementary materials.
>
> 4 & 5) We ran an additional experiment evaluating perplexity of all 1.2B variants at sequence lengths 512, 1024, and 2048 tokens. The key finding is that the relative degradation of Q–K=V vs QKV remains consistent across all sequence lengths (+8.2% at 512, +7.6% at 1024, +8.4% at 2048), confirming that the K=V constraint does not compound or worsen at longer contexts. The relative rankings across all variants are also stable across sequence lengths, mirroring the results in Table 10. This suggests that projection sharing imposes a fixed representational cost that does not interact negatively with context length — an important property for long-context deployment scenarios where KV cache pressure is most acute. Will add this analysis to the revision.
>
> 6) We don't have results for 7B+ models. For quality at larger scale, our 300M → 1.2B results provide a clear trend: Q–K=V degradation decreases from 3.1% to 2.48% as model size increases. This suggests larger models are more robust to projection constraints, consistent with the general finding that over-parameterized models tolerate architectural constraints better.
>
> Questions:
> Q1 (RoPE): Our LMs use learned absolute positional embeddings, not sinusoidal encodings. The 2D sinusoidal encodings in "+" variants are specific to vision/synthetic tasks to inject asymmetry — unrelated to LM positional encoding. RoPE is fully compatible with all projection-sharing variants — it rotates Q and K independently before attention, orthogonal to the K=V weight-sharing constraint which operates at the projection level. We expect our quality-efficiency tradeoffs to hold with RoPE.
> Q2: All models are trained from scratch on SlimPajama. The perplexity values reflect pretraining at ~8.85B tokens.
> Q3: Perplexity is evaluated on a held-out SlimPajama validation split not used during training, evaluated every 500 steps over 10M validation tokens.

---

> > ### Author Rebuttal · Reviewer_x9XV · 2026-04-03
> >
> > Thank you for the rebuttal and provided evaluations on the downstream tasks. Indeed, benchmark evaluations give a rich signal allowing to better assess the performance compared to ppl.
> > I am raising the score from 3 to 4.

---

### Official Review · Reviewer_9mhq · 2026-03-12

**Soundness:** 3
**Presentation:** 3
**Significance:** 2
**Originality:** 2
**Overall Recommendation:** 3
**Confidence:** 3

**Summary:**

This paper studies whether transformers really need separate query, key, and value projections in attention. The authors evaluate several weight-sharing variants, including sharing key and value, sharing query and key, and sharing all three, across synthetic reasoning tasks, vision benchmarks, and language modeling. Their main finding is that sharing key and value leads to only a small drop in performance in many settings, while also cutting KV-cache size in half. They also show that this approach is compatible with other efficiency methods such as GQA and MQA. Overall, the paper presents weight sharing in attention as a simple way to improve efficiency without greatly hurting performance.

**Compliance With Llm Reviewing Policy:**

Affirmed.

**Key Questions For Authors:**

- How do the authors distinguish this paper from closely related work, such as *Does Self-Attention Need Separate Weights in Transformers?* (Kowsher et al., 2025), and what should be considered the main novel contribution here?
- Can the authors provide a stronger theoretical or mechanistic justification for why the Q-K=V variant works relatively well, beyond the current empirical correlation / effective-rank analysis?
- For completeness, why was the Q=V sharing variant not evaluated? It seems like a natural additional ablation alongside K=V, Q=K, and Q=K=V.
- Can the authors fully specify the GPT-style language-modeling architecture and implementation details, including normalization, activation, positional encoding, tokenizer/vocabulary, and any other choices that could materially affect perplexity?
- Have the authors evaluated the “+” variants in causal language modeling, where directional asymmetry seems most important? If not, why not?

**Limitations:**

The paper does not sufficiently discuss empirical limitations. In particular, the results are limited to a specific set of tasks and a somewhat under-specified GPT-style setup, so the conclusions may not transfer cleanly to stronger modern LLM baselines.

**Strengths And Weaknesses:**

### Strengths

- The paper studies whether transformers really need three separate query, key, and value projections. This is a relevant efficiency question, and the paper explores it systematically via a broad empirical study across synthetic tasks, vision benchmarks, and language modeling, rather than as a narrow ablation.
- The Q ⁣− ⁣K=V variant performs fairly well. It preserves asymmetric attention and reduces KV-cache size by 50% at the cost of modest perplexity degradation in the language-modeling experiments (3.1% at 300M, 2.48% at 1.2B). This is a clean and practically relevant takeaway.
- In addition, the paper does a good job connecting projection sharing to existing techniques such as GQA and MQA. Showing that projection sharing and head sharing are complementary makes the work more practically grounded, and the combined variants reach very large cache reductions.
- Finally, the paper does not entirely hide negative results. It makes clear that Q=K=V performs poorly for causal language modeling, and that Q=K ⁣− ⁣V offers no cache benefit despite some competitive results elsewhere. This makes the overall presentation feel more balanced than a paper that only emphasizes the best case.

### Weaknesses

- My main concern is originality and related-work positioning. There is closely related recent work, *Does Self-Attention Need Separate Weights in Transformers?* (Kowsher et al., 2025), which asks a very similar high-level question and proposes a shared-weight self-attention mechanism for BERT using a single shared projection matrix with diagonal modulation. The present submission is broader in scope and more focused on projection-sharing variants plus KV-cache efficiency, but the overlap in core motivation is substantial enough that the novelty should be positioned more carefully.
- The paper presents a useful efficiency-performance tradeoff, but the practical case is somewhat mixed. The quality degradation of Q ⁣− ⁣K=V is modest, but not marginal. It may be acceptable given the 50% KV-cache reduction, but it is still worse than strong head-sharing baselines such as GQA and MQA on the reported language-modeling results.
- For the NLP experiments, the GPT-style architecture is somewhat under-specified. The paper reports model scale, head count, MLP size, optimizer, and training schedule, but does not clearly describe several architectural choices that can materially affect language-model performance, such as the normalization type, activation function, positional encoding, and other implementation details relevant to LM quality. This makes it harder to assess how representative the setup is relative to modern GPT-style baselines, and also weakens reproducibility.
- The treatment of the 2D positional-encoding “+” variants is somewhat under-explored. The mechanism is introduced as a way to inject asymmetry into symmetric attention, but it feels heuristic and adds extra compute cost. More importantly, while the paper emphasizes the importance of directional asymmetry, the role of the “+” variants in causal language modeling is not explored nearly as thoroughly as one might expect.
- Finally, some of the explanatory claims feel stronger than the evidence. The paper argues that Q ⁣− ⁣K=V works because keys and values occupy similar representational spaces, supported by correlation and effective-rank analysis. This is interesting and directionally plausible, but feels more like supporting intuition than a fully convincing mechanistic or theoretical explanation.

---

> ### Author Rebuttal · Authors · 2026-03-29
>
> Thanks for your comments.
>
> Weakness:
>
> 1) We were not aware of this work at submission time. Note that we have preliminary draft of our work, published in 2003, prior to Kowsher et al 2005. Both works are not peer reviewed. We share details with AC. While Kowsher et al. and our paper share the same high-level motivation — questioning whether separate Q, K, and V projections are necessary — the two works differ substantially in scope, methodology, and contribution. They focus on a single shared projection with diagonal modulation applied to BERT-style encoder models, essentially asking a binary question: can one projection replace three? Our work instead conducts a systematic study of the full space of pairwise and triple sharing strategies, evaluated across 12 diverse tasks spanning synthetic reasoning, computer vision, and autoregressive language modeling at both 300M and 1.2B parameter scales.
>
> Crucially, our central finding goes beyond the question of whether sharing is possible. We identify which sharing strategy is most practically valuable and why: the Q–K=V configuration is beneficial because it halves the KV cache during autoregressive generation — a benefit that no other two-projection variant provides. This inference-time memory reduction, and its complementarity with head-sharing methods like GQA and MQA, is the primary contribution of our work, and has no direct analog in Kowsher et al.
>
> 2) This comparison somewhat misframes the contribution. Q–K=V is not a competitor to GQA or MQA — it operates on an orthogonal axis and is designed to be stacked on top of them. The most relevant comparison is therefore not Q–K=V versus GQA in isolation, but what happens when they are combined: Q-GQA-4 achieves 87.5% cache reduction with only 3.9% degradation, and Q-MQA reaches 96.9% with 4.8% — figures neither technique could approach alone.
>
> Additionally, the 3.1% degradation at 300M scale narrows to 2.48% at 1.2B, suggesting the tradeoff becomes more favorable as model size increases, precisely where KV cache pressure matters most.
>
> 3) All 1.2B models share an identical backbone. Each model consists of 22 layers with embedding dimension 2048, 32 attention heads (head dimension 64), and FFN dimension 8192 with GELU activation. Layer normalization (ε=1e-5) is applied before each attention and FFN sublayer (pre-norm). Input and output embeddings are tied, with vocabulary size 50304 using the GPT-2 tokenizer. Positional information is encoded via learned absolute position embeddings with maximum sequence length 2048. All models are trained from scratch on SlimPajama for ~8.85B tokens using AdamW (lr=6e-5, min_lr=6e-6, β1=0.9, β2=0.95, weight decay=0.1, gradient clipping=1.0), with 1000-step linear warmup followed by cosine decay, residual dropout 0.1, bfloat16 mixed precision, and gradient accumulation of 36 steps across 8×A100 40GB GPUs. The only architectural difference across variants is the attention mechanism — other components are identical, ensuring a controlled comparison. Perplexity is evaluated on a held-out SlimPajama validation split not used during training, evaluated every 500 steps over 10M validation tokens.
>
> 4) The (X)+ variants were primarily motivated by vision and synthetic tasks, where symmetric attention is the main limitation and 2D positional structure is a natural fit. In those settings they provide consistent gains (Table 3).
> For causal language modeling, however, asymmetry is already enforced by the causal mask, which means the "+" mechanism addresses a problem that does not meaningfully exist in that setting. This is precisely why we did not prioritize these variants in NLP experiments — their added compute cost is difficult to justify when Q–K=V already preserves the asymmetric attention structure that language modeling requires.
>
> We agree the paper does not make this reasoning explicit. In the revision, we will clarify that the "+" variants are specifically targeted at non-causal settings, acknowledge their heuristic nature, and add a brief discussion of why they are not the recommended path for language modeling. We also agree that a more principled mechanism for asymmetry injection is a worthwhile direction for future work, and we will note this explicitly. We will soften the causal language in the revision.
>
> 5)  The cosine similarity and effective-rank analysis are observational — they establish that K and V are similar in trained models, but do not formally explain why the constraint is benign. We present them as consistent, empirically grounded intuition rather than a mechanistic proof.
>
> We note that the contrast with Q — which maintains systematically lower similarity to both K and V — provides a clean internal control that strengthens the directional argument. A theoretical characterization of when and why K=V incurs bounded performance loss remains an open problem. We will flag in the revised discussion.
>
> Key Question: For Q=V, please see our response to Reviewer nbTi #3

---

> > ### Author Rebuttal · Reviewer_9mhq · 2026-04-03
> >
> > The rebuttal helps clarify some implementation details and scope, but my main concerns are still not fully addressed. In particular, the novelty relative to closely related prior work remains unclear, and the justification for why the proposed sharing strategy works is still mainly empirical rather than mechanistic.

---

> > > ### Author Response · Authors · 2026-04-03
> > >
> > > Thanks again for the follow-up — we wanted to come back to your remaining concerns more directly.
> > >
> > > 1) On novelty relative to Kowsher et al.
> > >
> > > We think the framing in our rebuttal undersold how different the two papers actually are. Kowsher et al. ask whether a single shared projection can replace three in a BERT encoder — a binary question, one architecture, one domain, no inference-time implications. We're asking something different: across the full space of sharing strategies, which one is actually worth using, and why? The answer — Q–K=V — only emerges when you care about autoregressive generation, which BERT-style models don't do. The KV cache reduction, the GQA/MQA interaction, the downstream task results — none of that has an analog in their work. The motivation overlaps, but the papers diverge immediately after the opening question.
> > >
> > > Also, note that we have an early draft on this work published in Arxiv back in 2023 which was not cited by Kowsher et al. We can share details with AC and they may be able to share it with you.
> > >
> > >
> > > 2) On mechanistic justification
> > >
> > > Fair point that the cosine similarity and effective-rank results are observational. But we'd push back a
> > > little on "mainly empirical" as a criticism here — and we think there's a more intuitive way to see why K=V is a natural constraint. But first, it's worth noting that this criticism cuts both ways: the original three-projection QKV design itself carries no theoretical justification. It was motivated by intuition and the database analogy — queries retrieve information by matching keys to retrieve values — not derived from any formal necessity. The separation of K and V was an architectural choice, not a theorem.
> > >
> > > With that in mind, think of keys and values like entries in a dictionary: the key is the headline that tells you what information is stored, and the value is the content itself. They're not the same thing, but they describe the same underlying item — they live in the same semantic neighborhood by construction. Tying K and V is therefore less like a restriction and more like making explicit what the network already learns on its own — and it is no less theoretically grounded than the original design it modifies. Our cosine similarity (0.73), effective rank results (687 vs. 702), and Q as an internal control all bear this out consistently — if the analysis were noise, Q should look similar to K and V too, and it doesn't.
> > >
> > > Beyond the mechanistic story, we think this has real practical significance that's easy to understate. Fewer projections means lower memory footprint, fewer parameters to store and update, and reduced compute at every forward pass — properties that matter enormously in edge and resource-constrained deployment scenarios. The fact that expressivity holds up despite fewer projections isn't just a nice empirical result; it suggests the standard QKV parameterization may be somewhat over-specified for many tasks, and that Q–K=V sits in a sweet spot where representational capacity is preserved while the parameter budget is meaningfully reduced. We think that's a finding worth building on.
> > >
> > >
> > > 3) Further benchmarking (following from 2)
> > >
> > > We ran actual benchmarks. If our mechanistic explanation were wrong — if K=V were actually a costly constraint — you would expect the model to suffer badly. But it doesn't. Q–K=V loses only 0.41% accuracy across five downstream benchmarks and actually runs faster and uses less memory than QKV in every configuration we tested: 4.4–5.3% higher decode throughput, 4.3–5.0% lower per-token latency, and 6.5–6.9% less GPU memory during autoregressive generation on a single A100. The fact that it works this well is itself mechanistic evidence — a constraint that is genuinely costly doesn't disappear at inference time. We agree a formal proof would be stronger and have acknowledged that gap explicitly in the revised discussion.
> > >
> > > Hope this helps clarify — happy to discuss further if useful.

---

### Official Review · Reviewer_nbTi · 2026-03-13

**Soundness:** 3
**Presentation:** 3
**Significance:** 4
**Originality:** 4
**Overall Recommendation:** 5
**Confidence:** 3

**Summary:**

Transformers have three projections in their attention layer: Q, K, and V. The authors conduct a systematic study of various projection-sharing schemes and find that Q-K=V projections can perform on par with standard transformers. They also show some complementarity with head-sharing schemes like GQA and MQA.

**Compliance With Llm Reviewing Policy:**

Affirmed.

**Key Questions For Authors:**

1. Are there any other ways projections could be shared?
2. On K=V working, your interpretation seems to be that the V projection is less important. I find this interpretation unsatisfying. Why not conclude instead that K is less critical rather than V since this is the same as V=K?

**Limitations:**

yes

**Strengths And Weaknesses:**

Strengths
1. Genuinely interesting question
2. Complementarity with head sharing
3. Paper is direct and easy to follow
4. Experimental results for reasoning, vision and language

Weaknesses
1. The explanation for why K=V works is limited
2. What about Q=V

---

> ### Author Rebuttal · Authors · 2026-03-29
>
> Thanks for your review.
>
> 1)  The explanation for why K=V works is limited
>
> We agree this deserves a sharper explanation. K and V weight matrices naturally converge to similar representations during training, with a mean cosine similarity of 0.73 across layers and nearly identical effective ranks (687 vs. 702 out of 1024 dimensions). Forcing K=V is therefore not a strong constraint — it makes explicit what the network already learns on its own.
> Q, by contrast, maintains much lower similarity to both K (0.42) and V (0.31), reflecting its fundamentally different role in asymmetric addressing. This is precisely why Q=K is damaging — it ties two representations the network actively keeps apart — while K=V is largely free, tying two representations the network already pushes together.
>
> These findings suggest that the K=V constraint aligns with the network's natural representational tendency, imposing minimal additional restriction beyond what gradient descent already enforces. The Q=K constraint, by contrast, forcibly equates two representations that the network actively differentiates, explaining its disproportionate performance cost. We will highlight this empirically-grounded interpretation as the primary explanation in the revised manuscript.
>
> 2) Are there other ways projections could be shared?
>
> Beyond the three variants we study (Q=K–V, Q–K=V, Q=K=V), the full space of pairwise and triple sharing includes Q=V–K, partial sharing within multi-head attention (e.g., sharing projections across heads rather than within a head, which is distinct from GQA/MQA), and cross-layer projection sharing where attention matrices are reused across transformer layers. We also note that our 2D positional encoding augmentation (the (X)+ variants) represents a soft form of asymmetry injection that interacts with projection sharing in non-trivial ways. We will add a brief taxonomy of the broader projection-sharing design space in the revised paper to clarify where our three variants sit and what remains unexplored.
>
> 3) Q=V case
>
> Q=V is a natural ablation and we acknowledge its omission. Like Q–K=V, the Q=V variant preserves asymmetric attention since QK^T is computed using distinct Q and K projections — symmetry only breaks when Q and K are tied. However, Q=V offers no KV cache benefit during autoregressive generation: since K and V remain distinct tensors at inference time, both must be cached separately, leaving cache size identical to standard QKV. This makes Q=V less motivated than Q–K=V.
>
> More fundamentally, our motivation for tying K and V specifically — rather than Q and V — is grounded in the representational analysis we provide. Our CCA and cosine similarity results show that K and V occupy highly similar representational spaces across layers (mean cosine similarity 0.73, effective ranks 687 vs 702 out of 1024), while Q remains systematically more distant from both. This suggests that K and V naturally encode redundant information during training, making the K=V constraint a low-cost approximation of what the network already learns. Q=V, by contrast, ties two representations that do not exhibit this natural convergence — Q and V have lower mutual similarity — making it a less well-motivated constraint. We will add Q=V results in the revision alongside additional representational analysis to complete the full pairwise sharing design space.
>
> 4) Why conclude V is less critical rather than K?
>
> Our current framing (that the V projection is less critical) was motivated by the contrast with Q=K–V: when Q and K are merged, performance degrades due to loss of directional asymmetry in QK^T; but when K and V are merged, Q and K remain independent, so the asymmetric attention pattern is preserved. In this sense, what is lost in Q–K=V is the independent V projection, and the model recovers well — which we interpreted as evidence that V is the less critical of the two.
> One could say that K is flexible enough to absorb the V role without sacrificing much, which would highlight K's representational richness rather than V's redundancy. We will revise Section 4 to present both interpretations and clarify that our claim is specifically about the asymmetry requirement — that QK^T must remain non-symmetric for causal tasks — rather than a definitive claim about the relative importance of individual projections in isolation. This distinction will also make the contrast with Q=K–V cleaner.

---

> > ### Author Rebuttal · Reviewer_nbTi · 2026-04-04
> >
> > The rebuttal clarifies a lot and answers most of my questions. That said, while it is understandable that the Q=V case was not tried and that K=V specifically was tried due to the representational analysis, some results for the Q=V would have made the point even clearer. If Q=V happens to outperform K=V, the point breaks down, while the opposite would strengthen it. The additional representational analysis, as well as the Q=V results, will improve the paper a lot in my opinion.

---

### Decision · Program_Chairs · 2026-04-30

**Decision:**

Accept (regular)

**Comment:**

The paper propose a clear & systematic empirical study of projection sharing in transformer attention, identifying the Q K=V configuration as a practical trade-off that reduces KV cache size with modest performance degradation.

Reviewers agree the work is strong on experimental side and relevant. However some concerns remain about novelty relative to prior work and limited theoritical explanation.

Author's rebuttal addressed several issues but questions about theoretical grounding and positioning inside the literature persist.

Despite this remaining issue, this paper is solid enough to recommend acceptance.